# CD44 connects autophagy decline and ageing in the vascular endothelium

Lu Zhang [1,2] ✉, Peichang Yang[1,2], Jingxuan Chen[1], Zhiqiang Chen[1], Zhihui Liu[1], Gaoqing Feng[1], Fangfang Sha[1], Zirui Li[1], Zaoyi Xu[1], Yating Huang[1], Xiaotong Shi[1], Xuebiao Li[1], Jiatian Cui[1], Chenyi Zhang[1], Pei Fan[1], Liuqing Cui[1], Yunpeng Shen[1], Guangzhou Zhou[1], Hongjuan Jing[1] & Shiwei Ma[1]

The decline of endothelial autophagy is closely related to vascular senescence and disease, although the molecular mechanisms connecting these outcomes in vascular endothelial cells (VECs) remain unclear. Here, we identify a crucial role for CD44, a multifunctional adhesion molecule, in controlling autophagy and ageing in VECs. The CD44 intercellular domain (CD44ICD) negatively regulates autophagy by reducing PIK3R4 and PIK3C3 levels and disrupting STAT3-dependent PtdIns3K complexes. CD44 and its homologue clec-31 are increased in ageing vascular endothelium and *Caenorhabditis elegans*, respectively, suggesting that an age-dependent increase in CD44 induces autophagy decline and ageing phenotypes. Accordingly, CD44 knockdown ameliorates age-associated phenotypes in VECs. The endothelium-specific CD44ICD knock-in mouse is shorter-lived, with VECs exhibiting obvious premature ageing characteristics associated with decreased basal autophagy. Autophagy activation suppresses the premature ageing of human and mouse VECs overexpressing CD44ICD, function conserved in the CD44 homologue clec-31 in *C. elegans*. Our work describes a mechanism coordinated by CD44 function bridging autophagy decline and ageing.

Endothelial cells are critical components of blood vessels, function at the interface between circulating blood and tissues and play many important roles in maintaining the physiological functions of the body[1]. Vascular endothelial cell (VEC) senescence is a hallmark of vasculature ageing that predisposes individuals to the onset of various cardiovascular diseases (CVDs)[1–3]. An improved understanding of endothelial senescence is critical for developing effective strategies to reduce the risk of CVD and maintain long-term health. Autophagy is a housekeeping mechanism critical for cellular homeostasis[4]. Some evidence shows that endothelial autophagy declines with age, and interventions that act on autophagy activation reverse arterial ageing and ameliorate age-related endothelial dysfunction[5,6], establishing causal links between endothelial ageing and autophagy decline. However, the detailed molecular basis underlying the age-related

decline in endothelial autophagy is currently unclear, resulting in a lack of potential targets for clinical intervention.

CD44 is a family of single-pass type I transmembrane glycoproteins consisting of three major domains: extracellular domain (ECD), transmembrane domain (TMD), and intracellular domain (ICD)[7]. As the most common marker for the identification, isolation and enrichment of cancer stem cells (CSCs), CD44 plays important roles in CSC maintenance, cancer metastatic progression, and chemotherapy resistance development[8] and has thus attracted substantial attention in the field of cancer research. Although CD44 is expressed widely in most cell types, including normal tissues[8], our understanding of its physiological function is largely limited to its role in the regulation of hyaluronic acid (HA) metabolism[9]. However, the roles and regulatory mechanisms of CD44 in normal vascular physiology remain largely unexplored.

[1]College of Bioengineering, Henan University of Technology, Lianhua Street, Zhengzhou 450001, China. [2]These authors contributed equally: Lu Zhang, Peichang Yang. ✉e-mail: zhanglu@haut.edu.cn

A previous study showed a significant reduction in aortic lesions in CD44-null mice compared with CD44 heterozygotes and wild-type littermates[10]. The CD44 levels are significantly higher in diseased arterial tissues than in normal tissues[11]. CD44 is highly expressed in senescent or inflammatory cytokine-stimulated endothelial cells and is thought to be involved in endothelial cell injury and monocyte adhesion[12–14]. Despite these clues showing an intimate link between CD44 and endothelial ageing, definitive evidence showing that CD44 is directly involved in ageing and, if so, how CD44 regulates this process is lacking. Inspired by the emerging association between CD44 and some autophagy-related proteins[15,16], we hereby aimed to elucidate whether there is a direct link between CD44 and autophagy and to uncover the critical roles of CD44 in the age-related decline in autophagy in VECs.

In this work, we demonstrate that the level of CD44 in VECs increases with age, which reduces the levels of the core components of the PtdIns3K complex, PIK3R4 and PIK3C3, and inhibits PIK3C3 kinase activities. In addition, CD44ICD binds PIK3R4 and PIK3C3 via binding and activating STAT3. As a result, PIK3R4 and PIK3C3 are lost in the PtdIns3K complex, giving rise to autophagy decline and ageing of VECs. Furthermore, we demonstrate that autophagy activation suppresses the premature ageing of VECs overexpressing CD44ICD, function conserved in the CD44 homologue clec-31 in *C. elegans*. Our study provides definitive evidence and a molecular basis for a direct connection between autophagy and VEC ageing. These observations uncover a function of CD44 in autophagy and ageing and suggest it as a potential target for delaying blood vessels ageing.

## Results

### CD44 contributes to VEC ageing

We serially passaged HUVECs to induce replicative senescence in vitro. SA-β-gal activity and the expression of CDKN1A (p21) and CDKN2A (p16), which are markers of cellular senescence, increased as the number of passages increased (Fig. 1a, b). The expression of CD44 at both the mRNA and protein levels increased in human umbilical vein endothelial cells (HUVECs) upon replicative senescence (Fig. 1b, c). To investigate the changes in CD44 levels in VECs upon natural ageing in vivo, aortas from young, middle-aged and aged C57BL/6 mice were isolated and subjected to assessment of the CD44 quantities. Data from qRT-PCR and western blot analysis showed that the levels of CD44 in the aortas of C57BL/6 mice increased with age (Fig. 1d, e). Strong immunoreactivity for CD44 was detected in the aortic endothelium of aged mice but not in that of young animals (Fig. 1f, g), consistent with previous observations in rats[12], suggesting that endothelial CD44 is upregulated in the ageing process.

To evaluate the potential role of CD44 in VEC senescence, we transfected HUVECs with Lenti-shCD44 to knock down CD44 (Fig. 1i). At population doubling (PD)14-PD18, these cells exhibited less SA-β-gal activity and decreased levels of CDKN1A and CDKN2A compared with the nontargeting control cells (Fig. 1h, i). The CD44 gene undergoes alternative splicing, generating standard CD44 (CD44s) and variant CD44 (CD44v) isoforms[7]. CD44s is encoded by constant exons and widely expressed in both cancer and normal tissues, whereas CD44v differs from CD44s by the insertion of variant exons, and their distribution is more restricted[8]. Overexpression of CD44s, the only isoform expressed in HUVECs[17], induced premature senescence in young HUVECs at PD3-PD6 (Fig. 1j, k).

In a cohort of aged mice, ageing markers were significantly lower in the vascular endothelium of CD44-knockout (CD44-KO) mice than in that of wild-type (WT) mice (Fig. 1l–n). Age-mediated changes in the arterial structure were observed in aged WT mice, as indicated by an increased aortic wall thickness, loss of the structural integrity of the aorta elastic lamella, and an increased collagen content compared with those in young mice (Fig. 1o–q). In CD44-KO mice, these age-dependent alterations in the vessel wall appeared to be ameliorated

(Fig. 1o–q). A common feature of VEC senescence is decreased endothelial nitric oxide synthase (eNOS) activity and nitric oxide (NO) levels[2]. The aortic endothelial expression of phosphorylated eNOS (p-eNOS) and the serum NO levels were higher in aged CD44-KO mice than in aged WT mice (Fig. 1r, s). These results collectively point towards an important role for CD44 in VEC ageing.

### CD44 decreases VEC autophagy

Autophagy decline leads to VEC senescence[5,6,18]. The question raised here is whether CD44 regulates autophagy. To address this question, we first evaluated autophagic flux in the endothelium of WT and CD44-KO mice injected with chloroquine (CQ), an inhibitor of autophagosome clearance. CQ did not induce significant changes in the number of LC3B puncta or the accumulation of SQSTM1 (a well-characterized substrate of autophagy) in aortic endothelial cells of aged WT mice (Fig. 2a, b), consistent with the predicted age-related decline in the autophagic capacity. In contrast, CQ increased the number of LC3B puncta and the accumulation of SQSTM1 in the endothelium of aged CD44-KO mice (Fig. 2a, b), suggesting that knockout of CD44 attenuated the age-related autophagy decline in VECs. In young mice, the numbers of LC3B puncta and autophagic structures in the endothelium were greater in CD44-KO mice than in WT mice (Fig. S1a, S1b), whereas the level of SQSTM1 was not increased (Extended Data Fig. 1c), indicating that CD44 deficiency enhances basal autophagy in vivo. CQ increased the accumulation of LC3B and SQSTM1 in the endothelium of both young WT and CD44-KO mice (Fig. S1d, S1e). The level of LC3B in the endothelium was greater in young CD44-KO mice than in WT mice (Fig. S1a, S1b, S1c) with or without CQ (Fig. S1d), confirming that knockout of CD44 causes animals to maintain a high level of basal autophagy during the ageing process.

Autophagy was then examined in cultured HUVECs transfected with specific siRNAs targeting CD44. The knockdown of CD44 increased the number of endogenous LC3B puncta and LC3B-II levels in HUVECs, and these increases were further enhanced in the presence of the autophagy inhibitor bafilomycin A₁ (Fig. 2c, d), suggesting intact autophagic flux. In parallel, CD44 knockdown resulted in marked degradation of the autophagy substrate SQSTM1 (Fig. 2d). Consistent with this, an increased number of both autophagosomes (yellow dots) and autolysosomes (red-only dots) in CD44-silenced HUVECs was observed by punctate dual fluorescent mCherry-GFP-LC3B expression (Fig. 2e). Ultrastructural analysis revealed the presence of a large number of crescent-shaped phagophores in CD44-silenced HUVECs (Fig. 2f), supporting the concept that CD44 deficiency enhances basal autophagy in VECs. Conversely, CD44-overexpressing HUVECs had a decreased number of LC3B puncta, decreased levels of LC3B-II and an apparent accumulation of SQSTM1, which was not altered by bafilomycin A₁ (Fig. 2g, h), suggesting that CD44 overexpression decreases autophagic activity rather than promotes autophagosome degradation. Moreover, CD44-overexpressing HUVECs exhibited a decreased number of both autophagosomes and autolysosomes compared with control cells (Fig. 2i, j), collectively demonstrating that CD44 decreases autophagy in VECs.

### CD44 decreases autophagy via its ICD

Interestingly, in addition to CD44s, other CD44 isoforms, including v3 and v7, also decreased autophagy in HUVECs (Fig. S2a, S2b). Different CD44 isoforms exist due to the alternatively spliced exons inserted within the ECD[19], whereas the ICD is equal to all CD44 variants and acts as an intracellular signal transduction molecule[20,21]. Our observation that both CD44s and CD44v isoforms decreased autophagy led us to hypothesize that CD44 might regulate autophagy via its ICD. To test this idea, we generated CD44-deletion mutants lacking the ICD (CD44ΔICD), ECD (CD44ΔECD), both the ECD and TMD (CD44ICD), or both the ICD and TMD (CD44ECD) (Fig. 3a). We found that CD44ECD

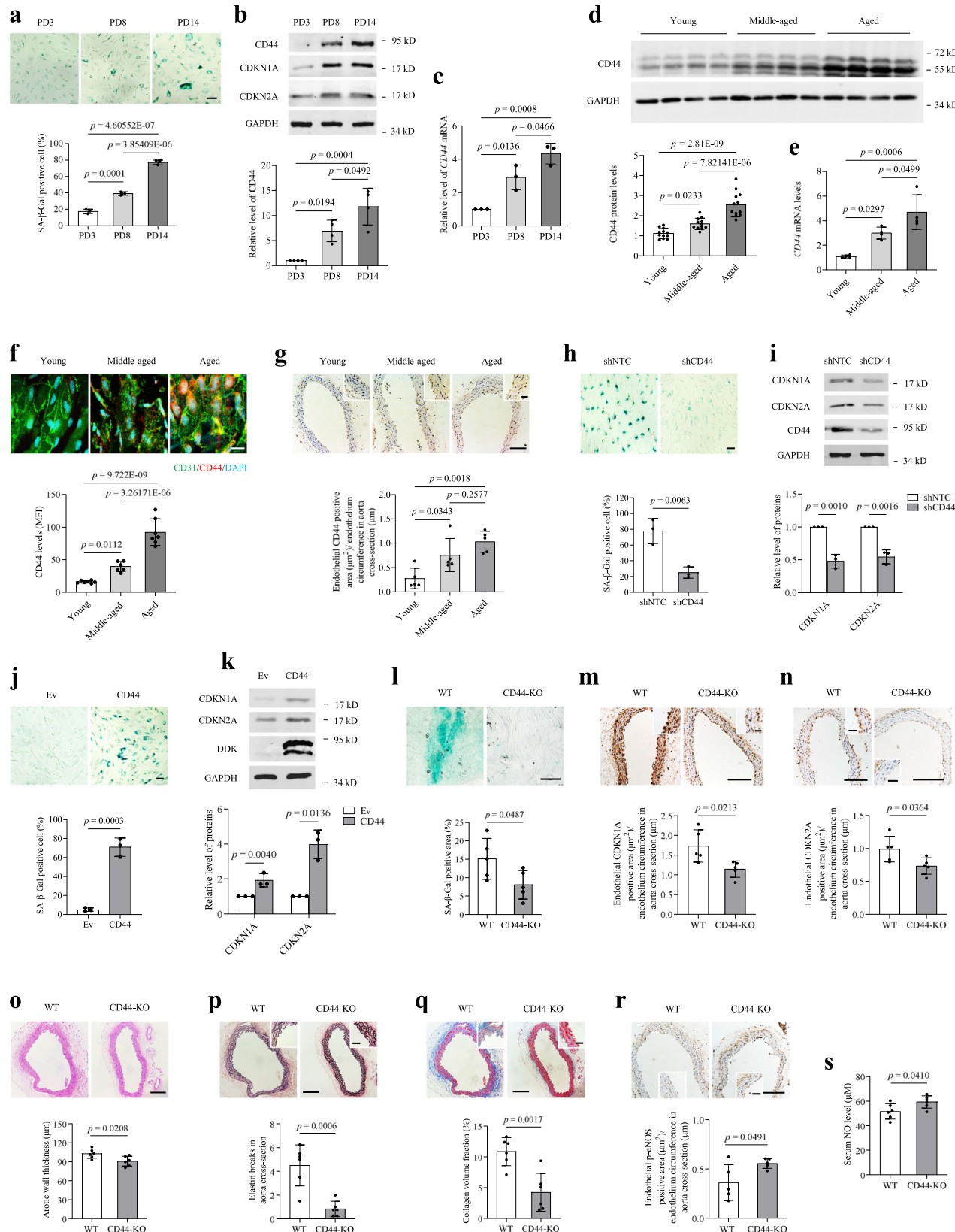

or CD44ΔICD did not reduce the LC3B-II levels or autophagosome-like structures or induce the accumulation of SQSTM1 in HUVECs, whereas CD44ICD or CD44ΔECD behaved similarly to full-length CD44 in reducing autophagy (Fig. 3b–d). CD44ICD-overexpressing HUVECs exhibited decreased numbers of both autophagosomes and autolysosomes (Fig. 3e). Treatment with bafilomycin A$_1$ indicated a bona

fide reduction in autophagy in CD44ICD-overexpressing HUVECs (Fig. 3f). To further prove the role of CD44ICD, we generated CD44-KO HUVECs using the CRISPR/Cas9 system. The level of basal autophagy in CD44-KO cells was higher than that in control cells, and this increase was fully rescued by reconstitution of full-length CD44s or CD44ICD (Fig. 3g), confirming that CD44 decreases autophagy via its ICD.

**Fig. 1 | CD44 exhibits age-related upregulation and contributes to VEC senescence. a** SA-β-gal staining in primary HUVECs with indicated population doubling (PD). Bar = 50 µm. **b, c** Western blot (**b**) and qRT-PCR (**c**) analysis of CD44, CDKN1A and CDKN2A in primary HUVECs with indicated PD. **d, e** Western blot (**d**) and qRT-PCR (**e**) analysis of aortas obtained from young (2–4 months), middled-aged (8–12 months), and aged (≥19 months) C57BL/6 mice. **f** *En face* staining for CD44 of aortas obtained from different age of C57BL/6 mice. MFI, mean fluorescence intensity. Bar = 20 µm. Green, CD31; Red, CD44. **g** Immunohistochemical staining for CD44 of aortas obtained from different age of C57BL/6 mice. **h, i** SA-β-gal staining (**h**) and western blot analysis of CD44, CDKN1A and CDKN2A (**i**) in HUVECs (PD14-PD18) expressing non-targeting control shRNA (shNTC) or shCD44. Bar = 30 µm. **j, k** SA-β-gal staining (**j**) and western blot analysis of CD44, CDKN1A and CDKN2A (**k**) in HUVECs (PD3-PD6) transduced with empty vector (Ev) or CD44.

**l, m, n, r** *En face* SA-β-gal staining (**l**) or immunohistochemical staining for CDKN1A (**m**), CDKN2A (**n**) and p-eNOS (**r**) of aortas obtained from WT (C57BL/6) or CD44-KO mice (16–19 months). **o–q** H&E staining (**o**), elastin staining (**p**) and Masson Tricolor staining (**q**) of aortic sections obtained from WT or CD44-KO mice (16–19 months). Bar = 200 µm. Bar in zoomed figure = 50 µm (**l–q**). **s** Serum NO level of WT or CD44-KO mice (16–19 months). For **d**, $n = 12$ mice per group. For **e**, $n = 4$ mice per group. For **f**, $n = 7$ (young or aged) or 6 (middle-aged) mice. For **g, l, m, n, r**, $n = 5$ mice per group. For **o, p, q, s**, $n = 6$ mice per group. Two (**d–g, l–s**) or three (**a, c, h–k**) or four (**b**) biologically independent experiments. Data are shown as mean ± s.d.; *P* values are derived from one-way ANOVA with Tukey's multiple comparisons test (**a–g**), Two-tailed unpaired Student's *t*-tests (**h–s**). Source data are provided as a Source data file.

To extend these observations to an in vivo situation, we generated an endothelium-specific CD44ICD-knock-in (CD44ICD*flox/flox*:Tek-Cre, CD44ICD$^{EC}$-KI) mouse model by crossing R26em CD44ICD-KI mice (CD44ICD*flox/flox*, *B6J.Cg-Gt(ROSA)26Sor$^{em16(CAG-CD44ICD)}$/J*) with endothelium-specific Tek-Cre mice (B6.Cg-Tg(Tek-cre)12Flv/J) (Fig. 3h, S3a–d). CD44ICD$^{EC}$-KI mice exhibited a decreased number of LC3 puncta and increased levels of SQSTM1 in the aortic endothelium compared with their littermate controls (CD44ICD*flox/flox*, hereafter denoted WT mice) (Fig. 3i, j). CQ did not significantly increase the accumulation of LC3B-II in the aorta of CD44ICD$^{EC}$-KI mice. The level of SQSTM1 in the aorta of CD44ICD$^{EC}$-KI mice was higher than that of WT mice (Fig. 3k).

To further identify the specific region responsible for the inhibitory effects of CD44ICD on autophagy, we generated three truncated CD44ICD mutants (Fig. S4a). Overexpression of truncated CD44ICD with 17 amino acid residues deleted from the N-terminus (CD44ICD_ΔN17) or truncated CD44ICD with 19 amino acid residues deleted from the C-terminus (CD44ICD_ΔC19) effectively reduced autophagy in HUVECs, whereas autophagy was not influenced by overexpressing truncated CD44ICD with 35 amino acid residues deleted from the N-terminus (CD44ICD_ΔN35) (Fig. S4b), suggesting that the 18-amino acid (S$^{305}$-A$^{322}$) stretch of CD44ICD is necessary to reduce autophagy. More interestingly, it has been demonstrated that the extensive phosphorylation of human CD44 in vivo occurs only on serine residues[22]. The 18 amino acid sequence of CD44ICD contains two serine residues, serine 305 (S305) and S316, which were mutated to alanine, singularly or in combination (Fig. S4c). The mutation of S305, but not S316, abolished the ability of CD44ICD to inhibit autophagy (Fig. S4d), suggesting that phosphorylation at S305 is needed for CD44ICD-induced autophagy reduction.

### CD44ICD release rather than nuclear translocation is necessary to reduce autophagy

CD44 undergoes intramembranous proteolysis by γ-secretase, resulting in the release of CD44ICD fragments[23]. To determine whether the release of CD44ICD is necessary for autophagy suppression, we generated an intracellular cleavage site deletion mutant (CD44Δ287-290) by eliminating four amino acids (I$^{287}$-N$^{290}$), which are essential regions for CD44 intracellular cleavage, as previously reported[23]. Unlike full-length CD44, CD44Δ287-290 failed to reduce autophagy in HUVECs (Fig. 4a–c). 24-Diamino-5-phenylthiazole (DAPT, GSI), a γ-secretase inhibitor that inhibits the release of CD44ICD, reversed the inhibitory effect of CD44 on autophagy but did not change the inhibitory effect of CD44ICD on autophagy (Fig. 4d), confirming that the release of CD44ICD is necessary to reduce autophagy.

The current understanding of CD44ICD signalling is limited to transcription regulation, which depends on its nuclear translocation[23–26]. Because the nuclear localization sequence (NLS) of CD44ICD is located in the N-terminal region[26], CD44ICD_ΔN17 is actually a mutant that cannot enter the nucleus. CD44ICD_ΔN17 still has the ability to reduce autophagy, suggesting that the CD44ICD-mediated inhibition of autophagy is independent of its nuclear translocation. To confirm this, we

generated a mutant by adding the myristoylation sequence (MGSSKSKPK) to the N-terminus of CD44ICD (Myr-CD44ICD), which fails to translocate to the nucleus[23]. Myr-CD44ICD also reduced autophagy in HUVECs (Fig. 4e–g), suggesting a nuclear translocation-independent role for CD44ICD in the regulation of VEC autophagy.

### CD44ICD reduces the level and activity of the PtdIns3K complex and alters the assembly state of the PtdIns3K complex via STAT3

Because CD44ICD decreases autophagy, we investigated whether the unc-51-like autophagy activating kinase 1 (ULK1) complex and class III phosphatidylinositol 3-kinase (PtdIns3K) complex, which play central roles in the initiation stage of autophagy[27], are targets of CD44. Although the knockdown of CD44 increased the levels of three core components of the ULK1 complex, ULK1, ATG13 and ATG101, no changes in the levels of the core components of the ULK1 complex were detected in CD44ICD-overexpressing HUVECs (Fig. S5a, Sb).

At least two distinct PtdIns3K complexes (complex I and II) are involved at different stages of mammalian autophagy. Both contain the core mechanisms of phosphoinositide-3-kinase regulatory subunit 4 (PIK3R4), phosphatidylinositol 3-kinase catalytic subunit type 3 (PIK3C3) and BECN1. Complex I (PtdIns3K-C1) is involved in autophagosome formation, which is defined by the presence of autophagy-related protein 14-like protein (Atg14L). Complex II (PtdIns3K-C2) replaces Atg14L with UVRAG, which is involved at later stages of autophagosome and endosome maturation[28]. The knockdown of CD44 increased the levels of PIK3R4 and PIK3C3, whereas the overexpression of CD44 or CD44ICD decreased their levels. The protein levels of BECN1, Atg14L and UVRAG did not vary with either CD44 or CD44ICD (Fig. 5a–d). In addition, an ELISA-based in vitro kinase reaction revealed that PtdIns3K kinase activity was significantly decreased after CD44 or CD44ICD overexpression compared with that in control cells (Fig. 5e), suggesting that CD44 reduced both the level and activity of the PtdIns3K complex.

We screened HUVECs transfected with two different siRNAs targeting CD44 for changes in gene expression using DNA oligonucleotide array-based genome-scale gene expression profiling. In the intersection of the two datasets, the signal transducer and activator of transcription (STAT) family attracted our attention. Four out of 7 STATs were downregulated (Fig. S6a), and the levels of STAT3 were consistent with the changes in CD44. That is, the knockdown or overexpression of CD44 caused their down- or upregulation, respectively (Fig. S6b, S6c). Congruently, a luciferase reporter assay revealed that overexpression of CD44 or CD44ICD enhanced the transcriptional activity of STAT3 in HUVECs (Fig. S6d). The knockdown of CD44 reduced the p-STAT3$^{Tyr705}$ levels, whereas the overexpression of CD44 or CD44ICD increased the p-STAT3$^{Tyr705}$ levels (Fig. S6e, S6f). The pharmacological inhibition or siRNA knockdown of STAT3 reversed the inhibitory effect of CD44ICD on autophagy (Fig. S6g, S6h), indicating that CD44ICD decreased autophagy via STAT3. Notably, the nonphosphorylatable STAT3 mutant STAT3$^{Y705F}$ failed to reduce autophagy (Fig. S6i),

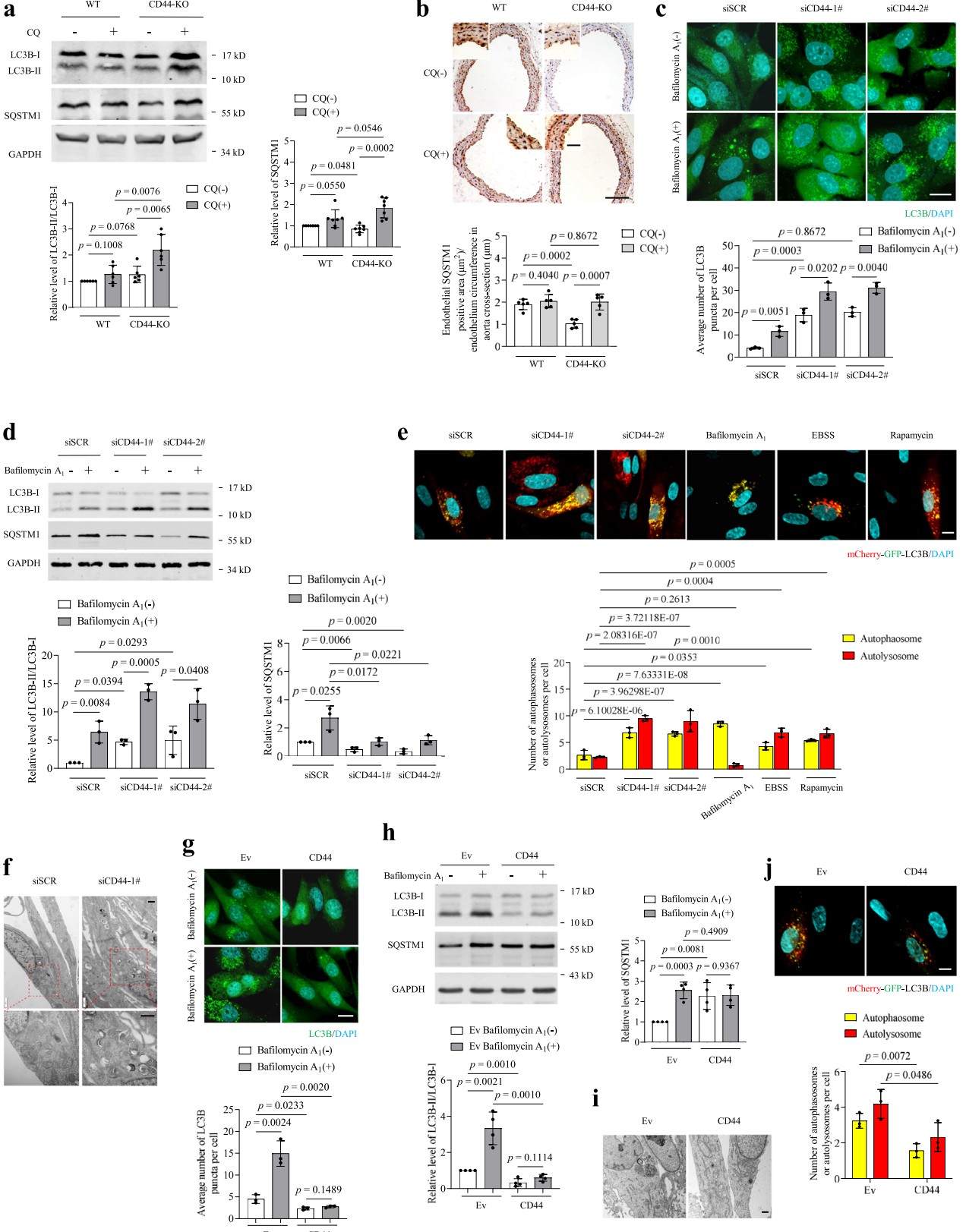

whereas the inhibitory effect of STAT3 on autophagy persisted when we employed a STAT3 variant that exclusively localized to the cytoplasm due to the addition of a nuclear export sequence (NES)[29] (Fig. S6j), suggesting that although STAT3 Tyr705 phosphorylation is needed, CD44 functions as a negative regulator of autophagy in a STAT3 transcription-independent manner.

More importantly, we observed that CD44ICD coimmunoprecipitated with STAT3, PIK3C3 and PIK3R4 but not with BECN1, Atg14L and UVRAG (Fig. S7), and this finding was confirmed by fluorescence microscopy (Fig. 6a), indicating that CD44ICD overexpression in VECs resulted in loss of PIK3C3 and PIK3R4 from the PtdIns3K complex. Using a yeast two-hybrid assay, we found that CD44ICD interacted

**Fig. 2 | CD44 negatively regulates autophagy. a, b** Western blot analysis of LC3B-II/LC3B-I and SQSTM1 (**a**) or immunohistochemical staining for SQSTM1 (**b**) of aortas obtained from WT or CD44-KO mice (16–19 months) treated with vehicle or chloroquine (CQ) by daily intraperitoneal injection (60 mg/kg/day) for 4 days. Bar = 200 μm. Bar in zoomed figure = 50 μm. **c, g** Fluorescence analysis of GFP-LC3B puncta in HUVECs (PD3-6) transfected with scrambled siRNA or CD44 siRNAs (**c**), or transduced with Ev or CD44ICD (**g**) in the presence or absence of bafilomycin A$_1$ (100 nM) for 6 h. **d, h** Western blot analysis of LC3B-II/LC3B-I and SQSTM1 in HUVECs (PD3-6) transfected with scrambled siRNA or CD44 siRNAs (**d**), or transduced with Ev or CD44 (**h**) in the presence or absence of bafilomycin A$_1$ (100 nM) for 6 h. **e, j** Fluorescence analysis of CD44 knockdown (**e**) or CD44ICD-overexpressing (**j**) HUVECs (PD3-PD6) transfected with mCherry-GFP-LC3B reporter. HUVECs treated with bafilomycin A$_1$ (100 nM) for 6 h were used as a positive control

for impaired autophagic flux. HUVECs incubated with EBSS for 3 h or treated with rapamycin (100 nM) for 6 h were used as positive controls for enhanced autophagy activity. Nuclei were stained with DAPI. Quantification of the average number of autolysosomes (GFP−mCherry+ puncta, red) and autophagosomes (GFP+mCherry+ puncta, yellow) per cell. Bar = 10 μm. **f, i** Ultrastructural analysis of HUVECs (PD3-6) transfected with scrambled siRNA or CD44 siRNAs (**f**), or transduced with Ev or CD44 (**i**). Bar = 500 nm. Boxed areas are enlarged below each image. For **a**, $n = 6$ (left bar graph) or 7 (right bar graph) mice per group. For **b**, $n = 5$ mice per group. Two (**a**, **b**) or three (**c–g**, **j**) or four (**h**) biologically independent experiments. Data are shown as mean ± s.d.; $P$ values are derived from Two-tailed unpaired Student's $t$-tests (**a**, **b**), one-way ANOVA with Dunnett's multiple comparisons test and Two-tailed unpaired Student's $t$-tests (**c–e**, **g**, **h**, **j**). Source data are provided as a Source data file.

directly with STAT3 but not with all the core components of PtdIns3K complexes. STAT3 interacted directly with CD44ICD and PIK3R4, a core component of PtdIns3K complexes (Fig. 6b). The direct interactions between CD44ICD and STAT3 and between STAT3 and PIK3R4 were confirmed by bimolecular fluorescence complementation (BiFC) and pull-down assays (Fig. 6c–e). Moreover, we built STAT3 and CD44ICD models using Alphafold2, and the protein–protein binding modes were then studied and analysed using GRAMM-X and the PDBePISA server. The affinity energy of the STAT3-CD44ICD complex is −7.2 kcal/mol, which indicates that these proteins bind tightly. The binding stability of the STAT3-CD44ICD complex is supported by the N-terminal 35 amino acids of CD44ICD, which formed multiple hydrogen bonds between STAT3 and CD44ICD (Fig. 6f). In the Co-IP analysis, CD44ICD missing 35 amino acids at the N-terminus (CD44ICD_ΔN35) failed to bind STAT3, PIK3R4 and PIK3C3 (Fig. 6g), indicating that the 35 amino acids are necessary for the combination of CD44ICD and STAT3. Interestingly, the inability of the CD44ICD_ΔN35 mutant to inhibit autophagy was observed in our data (Fig. S4b), which supports the viewpoint that CD44ICD blocks autophagy by interacting with STAT3. Based on these observations, we conclude that CD44ICD reduces the level and activity of the PtdIns3K complex and binds to PIK3R4 and PIK3C3 by directly interacting with STAT3, thereby disrupting the formation of the PtdIns3K complex and leading to reduced autophagy (Fig. 6h).

### CD44ICD mediates VEC senescence by reducing autophagy

We reasoned that if CD44 promotes ageing through its ICD, the endothelium of CD44ICD$^{EC}$-KI mice would exhibit premature ageing phenotypes. No significant difference in the heart rate or systolic and diastolic blood pressure (BP) was found between 8- to 10-month-old WT mice and CD44ICD$^{EC}$-KI mice (Fig. S8a–c), but the level of serum NO was lower in the CD44ICD$^{EC}$-KI mice (Fig. S8d). Senescence markers, including SA-β-gal staining and the CDKN1A and CDKN2A levels, were significantly higher in the aortic endothelium of 8- to 10-month-old CD44ICD$^{EC}$-KI mice than in age-matched control animals (Fig. 7a–c). Moreover, CD44ICD$^{EC}$-KI mice exhibited ageing-associated histological changes in the aorta that were not observed in WT mice (Fig. 7d–f). Compared with those of the WT animals, the aortas of CD44ICD$^{EC}$-KI mice had smaller lumen diameters during systole and diastole and also exhibited an increased blood flow velocity at the aortic arch (Fig. 7g–i). In the 7 different tissues evaluated, differences in the basal vascular permeability in the liver, lungs and skin were observed (Fig. S9), most likely as a consequence of impaired endothelial function. These results collectively suggested that CD44ICD caused premature ageing in VECs. Consistent with the in vivo findings, increased SA-β-gal staining and enhanced levels of CDKN1A and CDKN2A were observed in CD44ICD-overexpressing HUVECs (Fig. 7j, k). More than our expectation, the endothelial overexpression of CD44ICD significantly shortened the lifespan of mice (Fig. S10), suggesting that a

high level of endothelial CD44 may bring unimaginable harm to health. Of course, this needs further investigation.

We next sought to determine whether CD44ICD-mediated premature ageing is linked to the observed reduction in autophagy. Eight-month-old CD44ICD$^{EC}$-KI mice injected with rapamycin, an mTOR-dependent autophagy activator, or fed trehalose, an mTOR-independent autophagy activator, showed a decline in SA-β-gal staining as well as reduced CDKN1A and CDKN2A levels in the aortic endothelium (Fig. 7l–n, S11a–c). The activation of autophagy by rapamycin or trehalose yielded the same effects in vitro (Fig. 7o, p, S11d), suggesting that CD44 accelerates vascular endothelial senescence by reducing autophagy through its ICD.

### *C. elegans* CD44 homologue clec-31 is needed for connecting autophagy, ageing and longevity

To extend our observations across species, we searched for *C. elegans* CD44 homologues and found that *clec-31* exhibits sequence similarity to human CD44 and *Saccoglossus kowalevskii* CD44 (LOC100367193) (Fig. S12a, S12b). The *clec-31* gene localizes on chromosome 5, has seven exons and is predicted to have 368 amino acids. Clec-31 is the most likely candidate because its amino acid sequence is highly consistent with that of CD44 in other species. We sought to determine whether clec-31 functions similarly to CD44. GFP::LGG-1 displayed a diffuse distribution with occasional punctate structures in DA2123 nematodes fed bacteria expressing the L4440 vector, whereas GFP::LGG-1 formed a large number of punctate structures in DA2123 nematodes fed bacteria expressing dsRNA against clec-31 (Fig. 8a), suggesting that clec-31 regulates autophagy. In support of this notion, the knockdown of *clec-31* induced expression of a broad array of genes with distinct functions in the autophagic process (Fig. 8b), whereas overexpression of clec-31 (*clec-31* OE) suppressed their expression (Fig. 8c).

To explore whether clec-31 is involved in the regulation of ageing, we detected the level of clec-31 in *C. elegans* at different ages and found that clec-31 expression increased with ageing (Fig. 8d). The knockdown of clec-31 significantly extended the nematode lifespan (Fig. 8e), ameliorated the age-associated decline in the rates of body bending (Fig. 8f), pharyngeal pumping (Fig. 8g), and motor activity (Supplementary Movies 1 and 2), and reduced lipofuscin deposition (Fig. 8h). In contrast, the overexpression of clec-31 shortened the lifespan (Fig. 8i) and accelerated the age-associated degenerative changes in nematodes (Fig. 8j–l). In addition, although the total number of hatched eggs did not change significantly (Fig. S13a, S13b), the reproductive period of clec-31-knockdown nematodes was prolonged (Fig. S13c), whereas the reproductive period of clec-31-OE nematodes was shortened (Fig. S13d).

Next, we investigated whether clec-31 accelerates ageing and shortens the lifespan in *C. elegans* by reducing autophagy. The activation of autophagy in *clec-31*-OE nematodes by feeding bacteria expressing mTOR/let-363 dsRNA reversed the shortened

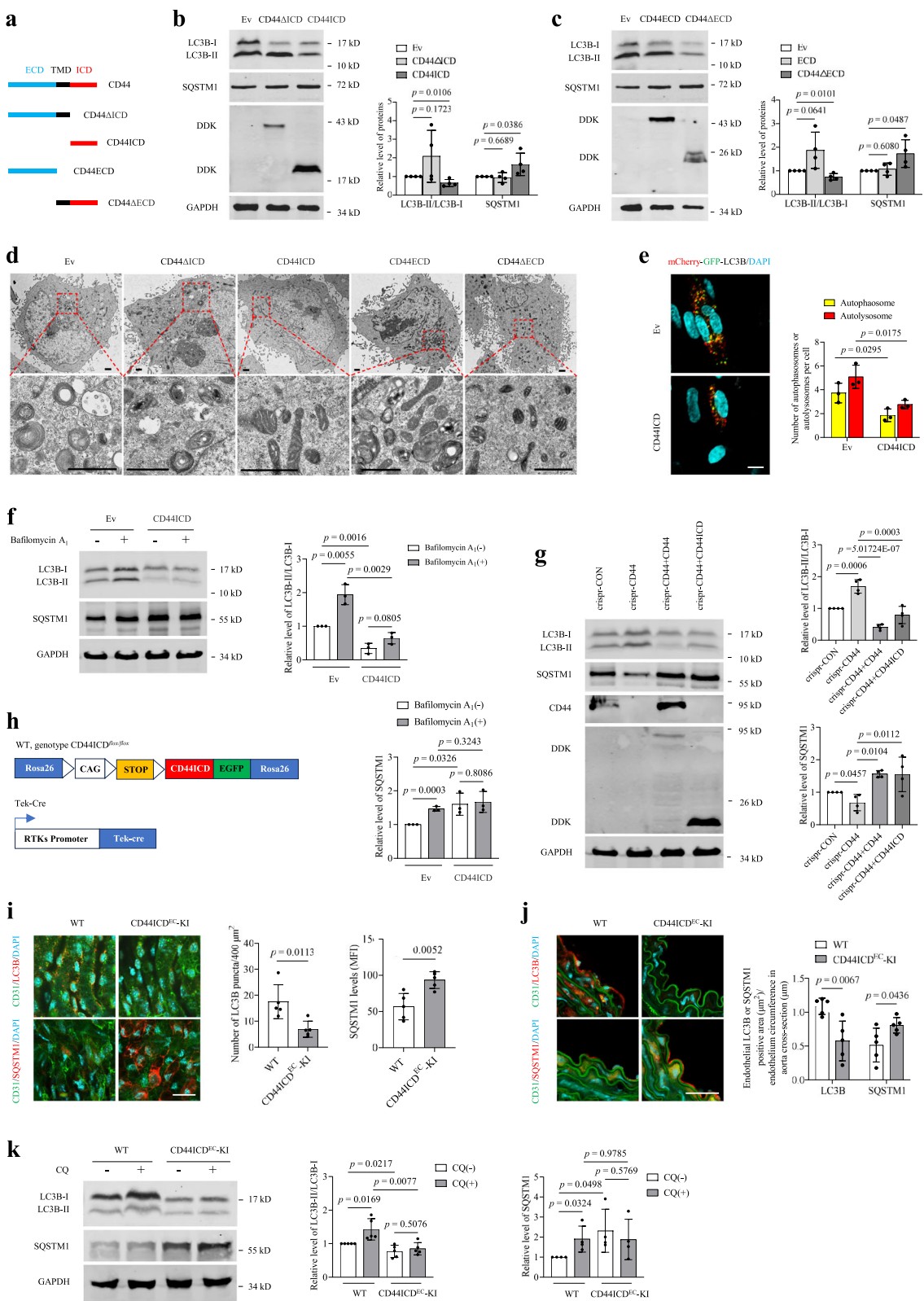

lifespan and age-associated phenotypes of *clec-31*-OE nematodes (Fig. 9a–d, S14a). Similar results were obtained from rapamycin-treated *clec-31*-OE nematodes (Fig. 9e–h, S14a). The genetic inhibition of autophagy by RNAi against *unc-51* or *vps34* suppressed the lifespan extension of *clec-31*-KD nematodes and rapamycin-treated *clec-31*-OE nematodes (Fig. 9i–k, S14b). Collectively, clec-31 appears to be a key regulator of senescence in nematodes through the

regulation of autophagy, and this regulation is functionally conserved via mammalian CD44.

## Discussion

Autophagy cleans cells by removing defective or aged organelles and macromolecules, thereby maintaining cellular homeostasis[30]. It is hardly surprising that this process is particularly important for

**Fig. 3 | CD44 reduced autophagy via its ICD. a** Schematic of the CD44 domain deletion mutants. **b,c**, Western blot analysis of LC3B-II/LC3B-I and SQSTM1 in HUVECs (PD3-PD6) transduced with Ev, CD44ΔICD and CD44ICD (**b**), or CD44ECD and CD44ΔECD (**c**). **d** Ultrastructural analysis of HUVECs (PD3-PD6) transduced with Ev, CD44ΔICD, CD44ICD, CD44ECD or CD44ΔECD. Boxed areas are enlarged below each image. Bar = 1 μm. **e** Fluorescence analysis of Ev or CD44ICD-overexpressing HUVECs (PD3-PD6) transfected with mCherry-GFP-LC3B reporter. Nuclei were stained with DAPI. Bar = 10 μm. **f** Western blot analysis of LC3B-II/LC3B-I and SQSTM1 in HUVECs (PD3-PD6) transduced with Ev or CD44ICD in the presence or absence of bafilomycin A₁ (100 nM) for 6 h. **g** Western blot analysis of LC3B-II/LC3B-I and SQSTM1 in CD44KO HUVECs, reconstituted CD44KO/CD44 HUVECs and reconstituted CD44KO/CD44ICD HUVECs. **h** Constructs used to generate the endothelium-specific CD44ICD-KI mouse model. ROSA26CD44ICDKI mice (genotype: CD44ICD*flox/flox*) were crossed with endothelial-specific receptor tyrosine kinase (Tek) Cre mice to generate

CD44ICD*EC*-KI mice (genotype: CD44ICD*flox/flox*:Tek-Cre). **i, j** *En face* staining (**i**) or immunohistochemical staining (**j**) for LC3B and SQSTM1 of the aortas from WT (CD44ICD*flox/flox*) and CD44ICD*EC*-KI (CD44ICD*flox/flox*:Tek-Cre) mice (8–10 months). Bar = 50 μm. **i** Green, CD31; Red, LC3B or SQSTM1. **j** Green, autofluorescence of the internal elastic lamina; Red, LC3B or SQSTM1. **k** Western blot analysis of LC3B-II/LC3B-I and SQSTM1 of aortas obtained from WT (CD44ICD*flox/flox*) and CD44ICD*EC*-KI (8–10 months) treated with vehicle or CQ by daily intraperitoneal injection (60 mg/kg/day) for 4 days. For **i, j**, *n* = 5 mice per group. For **k**, *n* = 5 (left bar graph) or 4 (right bar graph) mice per group. Two (**i–k**) or three (**d, e, f**) or four (**b, c, g**) biologically independent experiments. Data are shown as mean ± s.d.; *P* values are derived from one-way ANOVA with Dunnett's multiple comparisons test (**b, c**), Two-tailed unpaired Student's *t*-tests (**e, f, i–k**), one-way ANOVA with Dunnett's multiple comparisons test and Two-tailed unpaired Student's *t*-tests (**g**). Source data are provided as a Source data file.

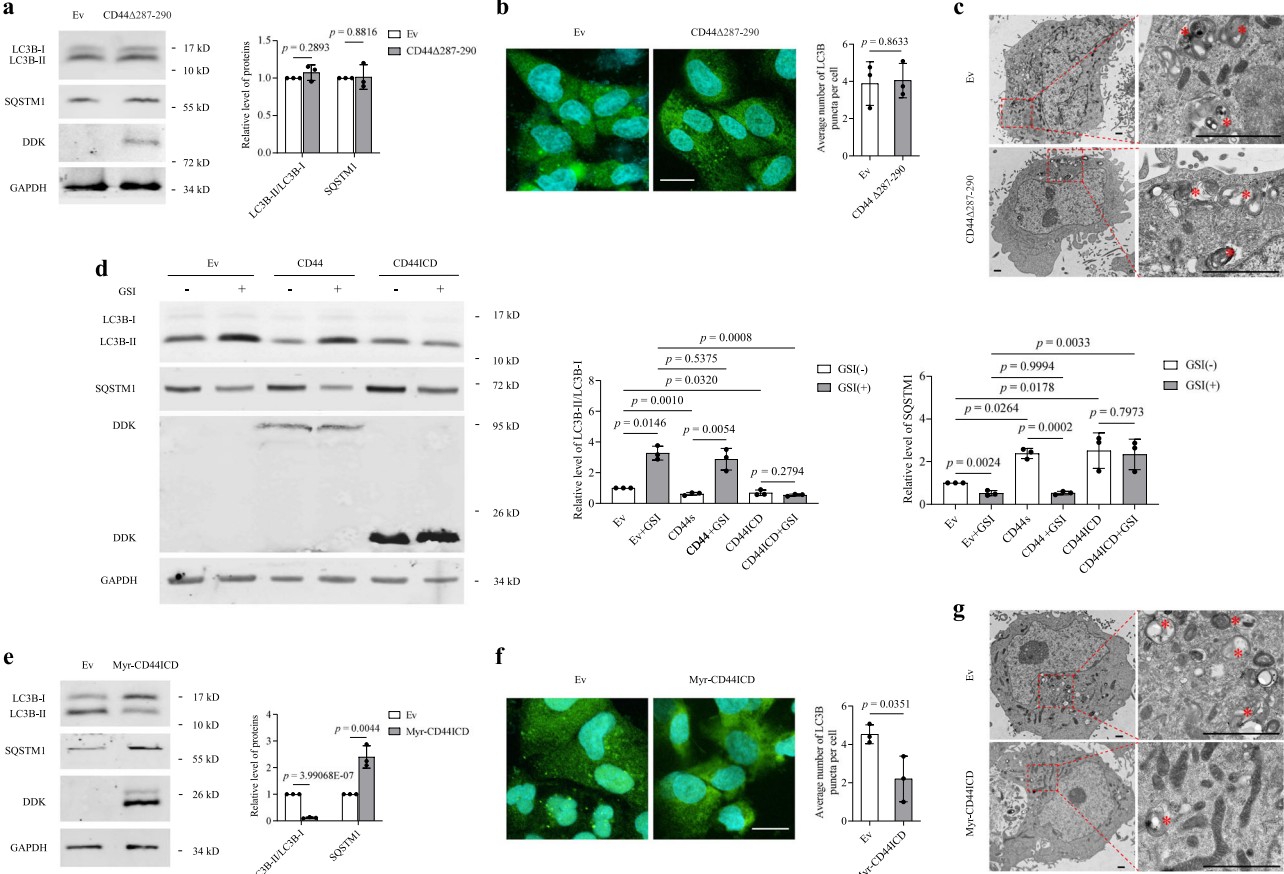

**Fig. 4 | CD44-mediated autophagy reduction requires CD44ICD release rather than nuclear translocation. a, e** Western blot analysis of LC3B-II/LC3B-I and SQSTM1 in HUVECs (PD3-PD6) transduced with Ev, CD44Δ287-290 (**a**) or Myr-CD44ICD (**e**). **b, f** Fluorescence analysis of GFP-LC3B puncta in HUVECs transduced with Ev, CD44Δ287-290 (**b**) or Myr-CD44ICD (**f**). Nuclei were stained with DAPI. Bar = 10 μm. **c, g** Ultrastructural analysis of HUVECs (PD3-PD6) transduced with Ev, CD44Δ287-290 (**c**) or Myr-CD44ICD (**g**). Boxed area of each image is enlarged to the

right. Asterisks indicate the autophagosomes/autolysosomes. Bar = 1 μm. **d** Western blot analysis of LC3B-II/LC3B-I and SQSTM1 in HUVECs (PD3-PD6) transduced with Ev, CD44 or CD44ICD in the presence or absence GSI (5 μM) for 3 h. Three biologically independent experiments. Data are shown as mean ± s.d.; *P* values are derived from Two-tailed unpaired Student's *t*-tests (**a, b, e, f**), one-way ANOVA with Dunnett's multiple comparisons test and Two-tailed unpaired Student's *t*-tests (**d**). Source data are provided as a Source data file.

maintaining the normal function of endothelial cells because they are continuously exposed to many circulating factors and pathogenic stimuli that predispose them to damage and are therefore particularly vulnerable to the accumulation of unfavourable proteins and damaged organelles. Increasing evidence suggests that autophagy decline contributes to endothelial cell ageing and the sharp rise in age-related CVD[5,6,18] but is poorly understood at the molecular level. To date, most CD44 research has focused on its involvement in cancer-associated signalling. In this study, we provide evidence showing an age-related

increase in the endothelial CD44 levels and describe the mechanism by which CD44 reduces autophagy and its functional impact on VEC senescence (Fig. S15).

Interestingly, CD44 was identified as one of the "senescence-induced cell adhesion genes", which is highly expressed in the aortic endothelium of old rats and responsible for enhancing the recruitment of monocytes to senescent endothelial cells[13,14], suggesting that CD44 is closely related to VEC senescence. However, it is unclear whether CD44 plays any role in the regulatory

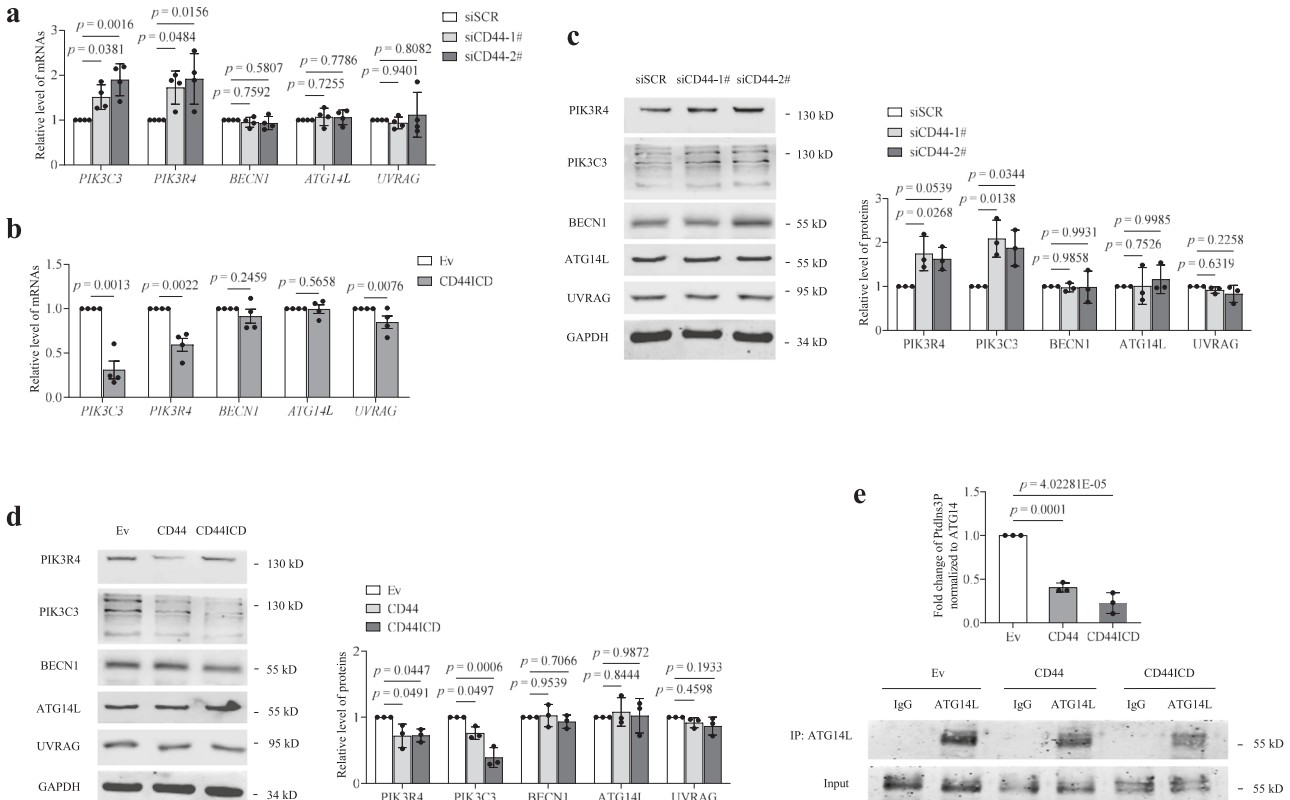

**Fig. 5 | CD44ICD reduces the level and activity of the PtdIns3K complex.**
**a**, **b** qRT-PCR analysis of the core proteins of the PtdIns3K complexes in HUVECs (PD3-PD6) transfected with scrambled siRNA and CD44 siRNAs (**a**), or transduced with Ev and CD44ICD (**b**). **c**, **d** Western blot analysis of the core proteins of the PtdIns3K complexes in HUVECs (PD3-PD6) transfected with scrambled siRNA or CD44 siRNAs (**c**), or transduced with Ev, CD44 and CD44ICD (**d**). **e** Endogenous ATG14L was immunoprecipitated from HUVECs and used for in vitro ELISA based PIK3C3 kinase activity assay. Different PtdIns3K complex components from Ev, CD44 or CD44ICD transfected HUVECs were immunoprecipitated using the ATG14L antibody. PIK3C3 activity was measured by analyzing PtdIns3P production in the ELISA assay described in the "Methods" section. The PtdIns3P fold change was calculated based on the concentration of PtdIns3P and was normalized to the amount of ATG14L used in the assay. Values are quantified minus IgG control reaction values. Three (**c**–**e**) or four (**a**, **b**) biologically independent experiments. Data are shown as mean ± s.d.; *P* values are derived from one-way ANOVA with Dunnett's multiple comparisons test (**a**, **c**–**e**), Two-tailed unpaired Student's *t*-tests (**b**). Source data are provided as a Source data file.

mechanism of ageing. Here, we showed that CD44ICD not only reduced the level and activity of the PtdIns3K complex but also disrupted the assembly of the PtdIns3K complex by directly interacting with STAT3, leading to autophagy decline and VEC senescence. Interestingly, although CD44ICD increases STAT3 transcriptional activity, it acts as a negative regulator of autophagy in a STAT3 transcription-independent manner. Notably, the nonphosphorylatable STAT3 mutant STAT3$^{Y705F}$ failed to repress autophagy in HUVECs. However, a previous report has shown that STAT3 inhibits autophagy via a cytoplasmic mechanism that does not involve the phosphorylation of Y705[29]. Obviously, these results are conflicting and may be explained by the different cell types used and conditions. Qin et al. reported that, unlike wild-type STAT3, overexpression of STAT3$^{Y705F}$ failed to repress autophagy in the human leukaemic monocyte lymphoma cell Line U937, providing evidence showing that p-STAT3 mediates the inhibition of starvation-induced autophagy by IL-6[31]. These findings support our observations. The CD44-mediated reduction in autophagy offers direct evidence and valuable insights into the causal relationship between age-dependent autophagy decline and vascular endothelial cell ageing. The fact that the endothelial overexpression of CD44ICD shortened the lifespan of animals is more than our expectation, indicating that its uncontrolled high expression may cause serious harm to health. Although at first surprising, this hypothesis is supported by several lines of evidence. First, accumulating evidence suggests that CD44 might be an early indicator of risk for atherogenesis[32–34]. CD44 alters gene expression in these areas prior to lesion development and even in the absence of any difference in the cholesterol levels[35]. Second, global-CD44-deletion mice exhibited a significantly lower lesion burden than WT mice with an atherosclerotic background[10]. Third, CD44 is predicted to be one of the key genes underlying the pathophysiological association between plaque instability and myocardial infarction progression[36]. Hence, it would be interesting to determine whether and how CD44-mediated autophagy decline contributes to age-related pathologies.

The overexpression of CD44 or CD44ICD reduces autophagy and promotes premature ageing of VECs, whereas the knockdown of CD44 rejuvenates ageing VECs, providing definitive evidence and a molecular basis for a direct connection between autophagy and endothelial cell ageing. Pharmacological and genetic evidence supports this connection because the activation of autophagy reversed the age-associated phenotypes induced by overexpression of CD44 or its homologue clec-31. Therefore, controlling the CD44 levels may provide a key means for avoiding the premature ageing of VECs, shedding light on potential therapeutic interventions to delay the ageing of blood vessels and even the whole body. Regulating the endothelial CD44 levels or intracellular hydrolysis may be a new direction for the design of autophagy regulators and compounds to maintain the health

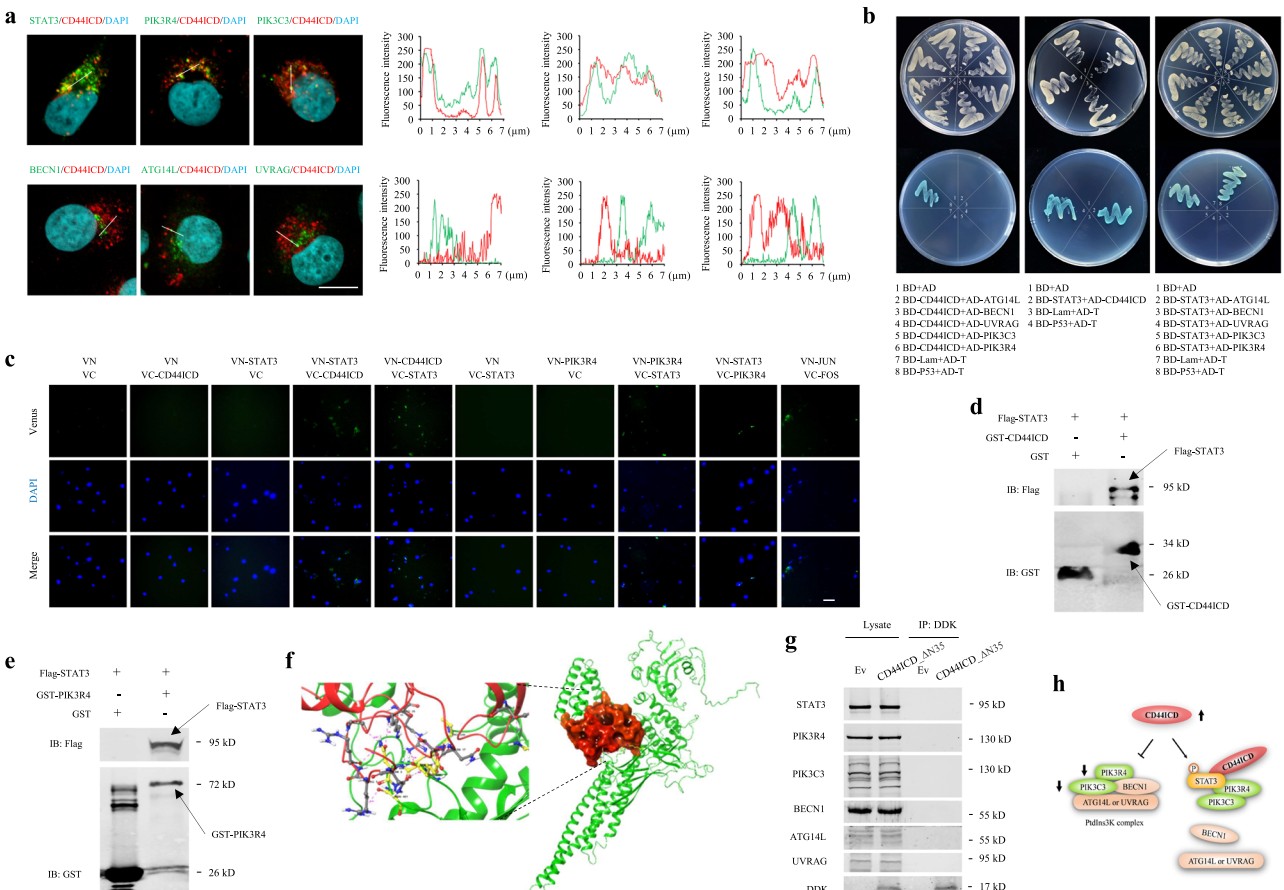

**Fig. 6 | CD44ICD alters the assembly state of PtdIns3K complex via STAT3.**
**a** Fluorescence analysis of the colocalization of CD44ICD with endogenous STAT3 and core proteins of the PtdIns3K complexes in HUVECs transduced with Ev or CD44ICD. Nuclei were stained with DAPI. Bar = 10 μm. The line scanned profiles at the right of confocal images show the distribution of fluorescence for each channel in the white line in the corresponding confocal images. **b** Reactivity of CD44ICD or STAT3 with PtdIns3K complex proteins, and STAT3 with CD44ICD in a yeast two-hybrid system. Yeast strain AH109 was co-transformed with a bait plasmid, BD-CD44ICD or BD-STAT3, and a prey plasmid, pGADT7-PtdIns3K complex proteins (AD-ATG14L, AD-BECN1, AD-UVRAG, AD-PIK3C3, and AD-PIK3R4) or pGADT7-CD44ICD, which encodes PtdIns3K complex proteins or CD44ICD fused to the Gal4 activation domain. Co-transformation of BD-Lam/AD-T and BD-P53/AD-T was used as negative and positive controls, respectively. **c** BiFC analysis of CD44ICD-STAT3 and STAT3-PIK3R4 interactions. HUVECs were transfected with indicated combinations of constructs. Co-transfection of VN-Jun and VC-Fos was used as positive control. Bar = 30 μm. **d**, **e** GST-pull-down of recombinant STAT3 with a GST-CD44ICD fusion protein (**d**) or a GST-P150 domain fusion protein (**e**). **f** The binding mode of full-sequence structures of STAT3 and CD44ICD. Colors indicate: STAT3, green; CD44ICD, red. The key residues interacting between STAT3 and CD44ICD are indicated as gray and yellow, respectively. Hydrogen bond is described by pink dash lines. **g** Western blots showing co-IP of CD44ICD_ΔN35 with endogenous STAT3 and core proteins of the PtdIns3K complexes in HUVECs transduced with Ev or CD44ICD_ΔN35 (DDK tag). Lysates, whole cell lysates; IP, immunoprecipitates. **h** Proposed schematic diagram of CD44ICD-mediated autophagy decline. CD44ICD suppresses the levels of PIK3C3 and PIK3R4 and the kinase activity of PIK3C3, activates STAT3, and disrupts the assembly of the PtdIns3K complex by interacting with STAT3. Three biologically independent experiments. Source data are provided as a Source data file.

of the cardiovascular system. Because ageing is at least partially due to impaired angiogenesis, our study seems to contradict previous findings showing that CD44 promotes pathological angiogenesis. In fact, it has been reported that CD44 deletion leads to reduced pathological angiogenesis without affecting normal angiogenesis[37]. We found that CD44 is expressed at low levels in normal young HUVECs and mouse vessels, which is consistent with the findings in the Human Protein Atlas database (https://www.proteinatlas.org): relatively low levels of CD44 on cardiomyocytes and VECs in normal human heart tissue. The expression of CD44 is upregulated in the myocardial infarcted and border zones[38]. Therefore, we realized that CD44 may not be necessary for physiological angiogenesis, and a recent study supports this notion[38]. In addition, there is evidence showing that in contrast to healthy blood vessels, pathological neovascularization engages pathways leading to p16 activation and ultimately culminating in VEC senescence. The selective ablation of dysfunctional neovascularization promotes regrowth of normal blood vessels[39]. These results led us to consider the possibility that elevated CD44 levels in VECs may promote pathological angiogenesis and VEC senescence, thereby preventing physiological vascular repair. The observation that elevated CD44 levels are associated with plaque destabilization supports this possibility[40]. Based on these observations, our results are not inconsistent with previous findings. The unanswered question in this study is the following: what is the mechanism driving the age-related increase of CD44 (and/or increased release of CD44ICD) in VECs? This question is worthy of further study in the future. It would also be interesting to determine whether CD44-mediated autophagy decline and ageing may also play a role in other cell types or tissues.

## Methods

All experiments were conducted in accordance with the ARRIVE guidelines and were carried out in accordance with the National Institutes of Health guide for the care and use of Laboratory animals (NIH Publications No. 8023, revised 1978) and approved by the Institutional Animal Care and Use Committee of Cyagen Biosciences Inc. (approval no. ACU22-035).

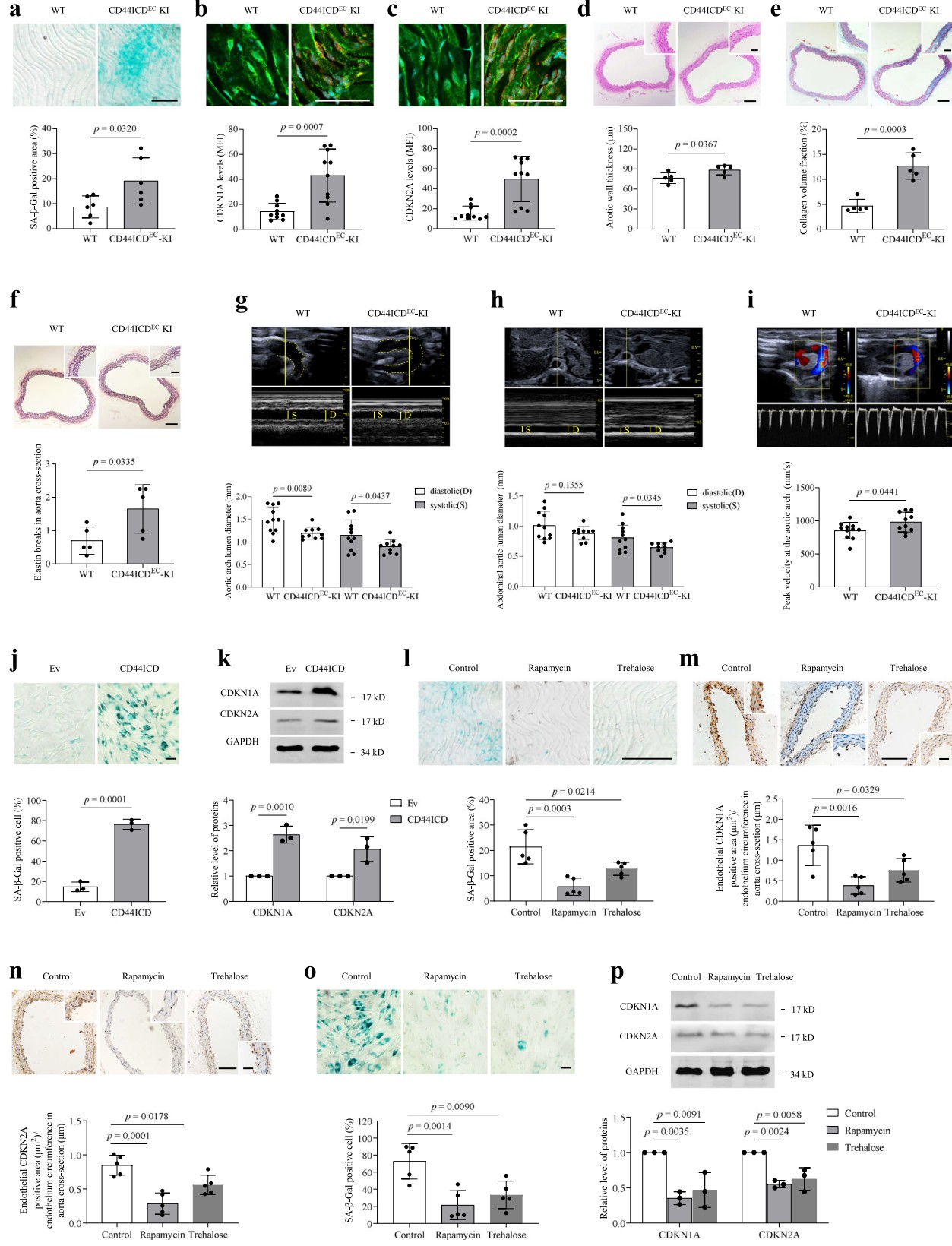

## Materials

The details of the antibodies, reagents, cells, animals, genechips, plasimds, and software used in this study are listed in Supplementary Table 1.

## Mice and treatments

CD44-KO mice (aged 8 weeks, B6.129(Cg)-Cd44 tm1Hbg/J) were purchased from Jackson Laboratories (Bar Harbour, Maine 04609, USA). C57BL/6 mice (aged 8 weeks) were purchased from Vital River

**Fig. 7 | Activation of autophagy reverses premature ageing of CD44ICD-overexpressing VECs. a–c** *En face* SA-β-gal staining (**a**) or immunohistochemical staining for CDKN1A (**b**) and CDKN2A (**c**) of aortas obtained from WT (CD44ICD*flox/flox*) and CD44ICD^EC-KI mice (8–10 months). Bar = 100 μm. Green, CD31; Red, CDKN1A and CDKN2A. **d–f** H&E staining (**h**), Masson Tricolor staining (**i**) or elastin staining (**j**) of aortic sections obtained from WT and CD44ICD^EC-KI mice (8–10 months). Bar = 150 μm. Bar in zoomed figure = 50 μm. **g–i** Echocardiography assessment of systolic (S) and diastolic (D) aortic arch lumen diameter (**j**) and abdominal aorta lumen diameter (**h**), and peak velocity of blood flow at the aortic arch (**i**) in WT and CD44ICD^EC-KI mice (14–17 months). **j, k** SA-β-gal staining (**j**) and western blot analysis of CDKN1A and CDKN2A (**k**) in HUVECs (PD3-PD6) transduced with Ev or CD44ICD. Bar = 30 μm. **l, m, n** *En face* SA-β-gal staining (**l**) or immunohistochemical staining for CDKN1A (**m**) and CDKN2A (**n**) of aortas from CD44ICD^EC-KI mice (8 months) received intraperitoneal injection of vehicle and rapamycin once every 2 days for a total of 6 injections (4 mg/kg/day), or fed with trehalose for 25 days (administration of 2% trehalose in drinking water). Bar = 150 μm. Bar in zoomed figure = 50 μm. **o, p** SA-β-gal staining (**o**) and western blot analysis of CDKN1A and CDKN2A (**p**) in CD44ICD-overexpressing HUVECs (PD2) treated with rapamycin (1 nM) or trehalose (100 μM) for 15 days. Bar = 30 μm. For **a**, $n = 6$ mice per group. For **b, c**, $n = 10$ mice per group. For **d, e, f, l, m, n**, $n = 5$ mice per group. For **g, h, i**, $n = 11$ (WT) mice, $n = 10$ (CD44ICD^EC-KI) mice. Two (**a–i, l–n**) or three (**j, k, p**) or four (**o**) biologically independent experiments. Data are shown as mean ± s.d.; *P* values are derived from Two-tailed unpaired Student's *t*-tests (**a–k**), one-way ANOVA with Dunnett's multiple comparisons test (**l–p**). Source data are provided as a Source data file.

Laboratory Animal Technology Co., Ltd. (Beijing, China). Tek-Cre mice (aged 10 weeks, B6.Cg-Tg(Tek-cre)12Flv/J) were purchased from Cyagen Biosciences Inc. (Suzhou, China). CD44ICD-KI floxed (CD44ICD*flox/flox*) mice were generated via CRISPR/Cas9-mediated genome engineering by Cyagen Biosciences Inc. CD44ICD*flox/flox* mice were crossed with Tek-Cre mice to produce CD44ICD^EC-KI (CD44ICD*flox/flox*:Tek-Cre) mice with CD44ICD-specific overexpression in the endothelium. The resultant CD44ICD^EC-KI mice were crossed with CD44ICD*flox/flox* mice, and the offspring were genotyped by PCR to identify CD44ICD^EC-KI mice and CD44ICD*flox/flox* mice as WT controls. All mice used in this study were maintained on a C57BL/6 J background. The animals were provided ad libitum access to water and standard laboratory chow (Laboratory Animal Center from the School of Medical Sciences of Zhengzhou University, Henan Province, China) and were housed on a 12 h light/dark cycle from 8 A.M. to 8 P.M. in a temperature and humidity-controlled environment (25 °C and 40–60%, respectively).

For CQ treatment, mice were treated with either vehicle or CQ (60 mg/kg/day) by daily intraperitoneal injection for 4 days. For rapamycin treatment, mice received an intraperitoneal injection of vehicle or rapamycin (4 mg/kg) once every 2 days for a total of 6 injections. For trehalose treatment, 2% trehalose was administered to the drinking water of the mice for 25 days. For survival analysis, the mice were monitored over the course of their natural lives. For other experiments, the mice were euthanized, and the blood and aorta were collected for subsequent research.

## Cell culture and treatments
HUVECs were isolated from the umbilical veins of the umbilical cords of healthy pregnant women by collagenase digestion (0.25 mg/ml) in our lab. Umbilical cord, as medical waste, bears no ethical consideration against any laws. The cells were cultured on gelatin-coated plastic dishes in Endothelial Cell Medium (ECM) containing 5% fetal bovine serum (FBS), endothelial cell growth supplement (1X), and 100 U/ml penicillin-streptomycin in a humidified incubator at 37 °C with 5% $CO_2$. The identity of the HUVECs was confirmed by their cobblestone morphology and strong positive immunoreactivity to CD31. The cells were subjected to serial passaging to induce replicative senescence.

For bafilomycin A$_1$ treatment, HUVECs were incubated in complete media supplemented with 100 nM of bafilomycin A$_1$ for 6 h. For Earle's balanced salt solution (EBSS) treatment, HUVECs were cultured in EBSS for 3 h. For GSI treatment, HUVECs were incubated in complete media supplemented with 5 μM of GSI for 3 h. For rapamycin treatment, HUVECs were incubated in complete media supplemented with 100 nM rapamycin for 6 h or 1 nM rapamycin for 15 days. For trehalose treatment, HUVECs were incubated in complete media supplemented with 100 μM trehalose for 15 days. The medium containing 1 nM rapamycin or 100 μM trehalose was changed every 2 days.

## Western blot analysis
After treatment, protein extraction was washed and lysed in lysis buffer containing 25 mM Tris-HCl (pH 6.8), 2% SDS, 6% glycerol, 2 mM PMSF, 1% 2-mercaptoethanol, 0.02% bromophenol blue and a protease inhibitor cocktail. The lysates were boiled for 15 min and immunoblotting were performed using standard methods. The nitrocellulose membranes were blotted with IRDye 800 labeled secondary antibodies, then the blots were scanned using a LI-COR Odyssey system (LI-COR Biosciences, Cambridge, UK). The images were analysed using Odyssey Application Software to obtain the integrated fluorescence intensities.

## Quantitative real-time PCR (qRT–PCR)
Total RNA was prepared from HUVECs and mouse tissues using the RaPure Total RNA Kit, and total RNA was extracted from *C. elegans* using the NGzol RNA Uptake Kit according to the manufacturer's instructions. RNA (2 μg) was utilized to perform reverse transcription using a HiScript II RT SuperMix for qPCR Kit. qRT–PCR was conducted using ChamQ Universal SYBR qPCR Master Mix on an Eppendorf PCR System (Eppendorf AG, Hamburg, Germany). For *C. elegans*, all assays were performed with 1000 animals per sample. The ΔΔCT method was used to calculate the relative expression of the tested genes normalized to the level of mammalian GAPDH or *C. elegans actin-1*. The primers used for qRT–PCR are listed in Supplementary Table 2.

## Immunostaining and histologic analysis
For immunofluorescence, HUVECs were seeded on coverslips and treated in 24-well plates. The cells were then fixed in 4% paraformaldehyde for 30 min at 4 °C. The aortas were fixed in 4% paraformaldehyde overnight. Following incubation with a sucrose gradient, the aortas were frozen in optimal cutting temperature (OCT) embedding medium (Tissue-Tek) and cut into 5-μm sections. For *en face* immunofluorescence of the aorta, the small rings of the aortas were cut and opened at the inner curvature. For immunohistochemistry, the aortas were fixed in 4% paraformaldehyde, subjected to gradient dehydration with different concentrations of alcohol, embedded in paraffin and cut into 4-μm sections with an ultrathin semiautomatic microtome (RM2016, Leica, Wetzlar, Germany). The samples were then permeabilized in 0.1% Triton-X, blocked with 5% normal serum from the same species as the secondary antibody and 2% BSA, and stained with the indicated antibodies in the figures. Immunofluorescence images were acquired under a confocal laser-scanning microscope (Olympus FLUOVIEW FV3000). Immunohistochemistry images were obtained using a microscope (DM IL LED Fluo, Leica Microsystems CMS GmbH, Wetzlar, Germany). All images were measured using ImageJ (National Institutes of Health) software.

For aortic wall thickness, collagen deposition and elastin analyses, the sections were stained with an H&E Staining Kit, Masson Tricolour Staining Kit, and Resorcinol staining solution kit, respectively,

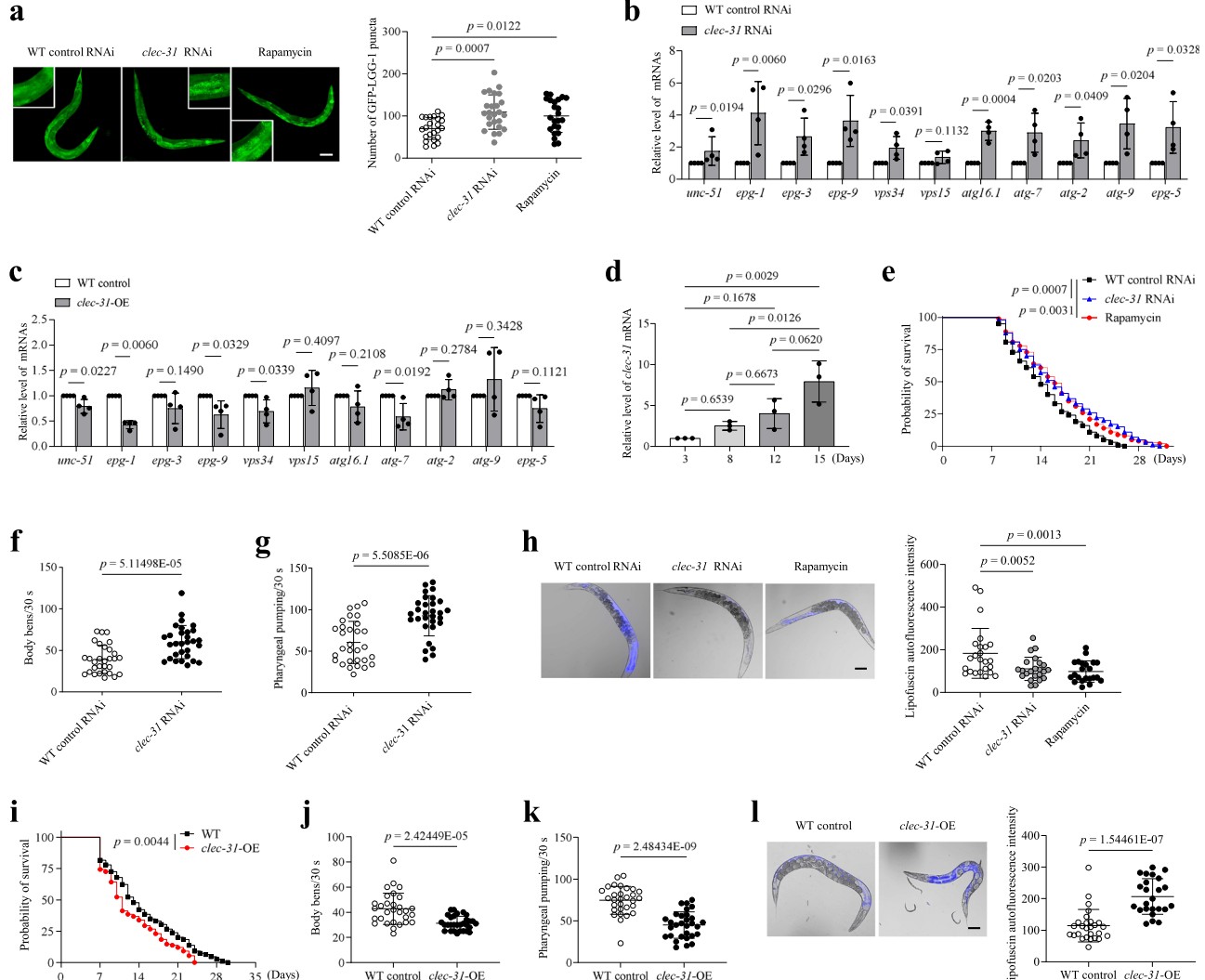

**Fig. 8 | *C. elegans* CD44 homologue clec-31 regulates autophagy, ageing and longevity. a** Detection of GFP::LGG1 puncta in nematodes fed RNAi bacteria targeting empty vector or *clec-31*. Nematodes that were treated with rapamycin (100 μM) were used as a positive control. Bar = 100 μm. Bar in zoomed figure = 50 μm. **b, c** qRT-PCR analysis of a panel of autophagy-related genes in nematodes fed RNAi bacteria targeting empty vector or *clec-31* (**b**) or in *clec-31*-overexpressing (OE) nematodes (**c**). **d** qRT-PCR analysis of clec-31 in nematodes aged different days as indicated in the figure. **e, i** Survival curves of (**e**) nematodes fed RNAi bacteria targeting empty vector, *clec-31* or treated with rapamycin as a positive control, (**i**) WT and *clec-31*-OE nematodes. All lines were raised and maintained at 20 °C. **f, j** The number of body bends monitored during a 30 s interval in (**f**) nematodes fed RNAi bacteria targeting empty vector or *clec-31*, **j** WT and *clec-31*-OE nematodes. **g, k** The number of pumps monitored during a 30 s interval in (**g**) nematodes fed RNAi

bacteria targeting empty vector or *clec-31*, (**k**) WT and *clec-31*-OE nematodes. **h, l** Detection of lipofuscin autofluorescence in (**h**) nematodes fed RNAi bacteria targeting empty vector or *clec-31*, **l** WT and *clec-31*-OE nematodes. For **a**, *n* = 23 (WT control RNAi or Rapamycin) nematodes, *n* = 25 (*clec-31* RNAi) nematodes. For **e**, *n* = 100 nematodes per group. For **f, g, j, k**, *n* = 30 nematodes per group. For **h**, *n* = 24 (WT control RNAi) nematodes, *n* = 23 (*clec-31* RNAi) nematodes, *n* = 22 (Rapamycin) nematodes. For **i**, *n* = 109 nematodes per group. For **l**, *n* = 27 (WT control) nematodes, *n* = 24 (*clec-31*-OE) nematodes. Three (**a, d, e–l**) or four (**b, c**) biologically independent experiments. Data are shown as mean ± s.d.; *P* values are derived from one-way ANOVA with Dunnett's multiple comparisons test (**a, h**), Two-tailed unpaired Student's *t*-tests (**b, c, f, g, j–l**), one-way ANOVA with Tukey's multiple comparisons test (**d**), Log-rank (Mantel-Cox) test (**e, i**). Source data are provided as a Source data file.

according to the manufacturer's protocols. The aortic wall thickness and collagen deposition were quantified using Image-Pro Plus software (Media Cybernetics). Each data point represents the average number of elastin breaks in the aortic cross-section of each animal.

### SA-β-Gal staining
The detection of SA-β-gal activity was conducted using the Senescence β-Gal Staining Kit according to the manufacturer's protocol. After treatment, HUVECs were fixed at room temperature for 10 min and incubated with an X-gal staining solution at 37 °C overnight. The percentage of blue-stained cells was calculated under a microscope. For *en face* β-Gal staining, the aortas were incubated with an X-gal staining solution at 37 °C for 48 h and mounted *en face* on glass slides. The

images were taken under a microscope, and blue-stained areas were measured using ImageJ software.

### Plasmids
The CD44, CD44v3, CD44v7, CD44ECD, CD44ICD, CD44ΔICD, CD44ΔECD, CD44Δ287-290, CD44ICD_ΔN17, CD44ICD_ΔC19, CD44ICD_ΔN35, CD44ICD_Mut1, CD44ICD_Mut2, and CD44ICD_Mut3 sequences were cloned and inserted into the pCMV-3xFlag vector. The Myr-CD44ICD construct was prepared using PCR to add sequences encoding the myristoylation amino acids to the N end of CD44ICD and cloning the fragment into pCMV-3xFlag.

For the GST pull-down assay, the sequence of CD44ICD and a partial sequence of PIK3R4 were cloned and inserted into pGEX-4T-3

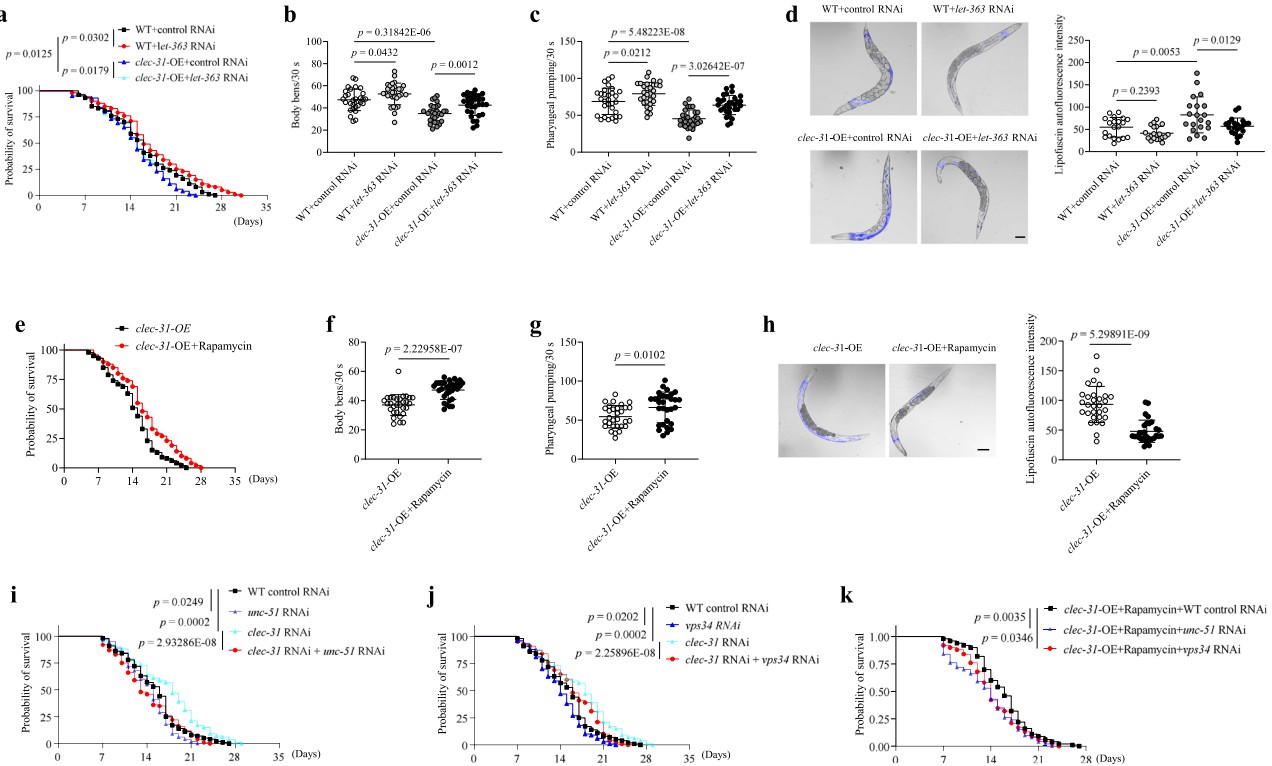

**Fig. 9 | clec-31 regulates senescence and longevity through regulating autophagy. a**, **e** Survival curves of (**a**) WT and *clec-31*-OE nematodes fed RNAi bacteria targeting empty vector or *let-363*, **e** *clec-31*-OE nematodes treated with or without rapamycin from day 1 of adulthood. All lines were raised and maintained at 20 °C. **b**, **f** The number of body bends monitored during a 30 s interval in (**b**) WT and *clec-31*-OE nematodes fed RNAi bacteria targeting empty vector or *let-363*, **f** *clec-31*-OE nematodes treated with or without rapamycin. **c**, **g** The number of pumps monitored during a 30 s interval in (**c**) WT and *clec-31*-OE nematodes fed RNAi bacteria targeting empty vector or *let-363*, **g** *clec-31*-OE nematodes treated with or without rapamycin. **d**, **h** Detection of lipofuscin autofluorescence in (**d**) WT and *clec-31*-OE nematodes fed RNAi bacteria targeting empty vector or *let-363*, **h** *clec-31*-OE nematodes treated with or without rapamycin. Bar = 100 μm. **i**–**k** Survival curves of (**i**) nematodes fed RNAi bacteria targeting empty vector, *clec-31* or *unc-51*, alone or

in combination, **j** nematodes fed RNAi bacteria targeting empty vector, *clec-31* or *ups34*, alone or in combination, **k** *clec-31*-OE nematodes fed RNAi bacteria targeting empty vector, *unc-51* or *vps34* in the presence of rapamycin. All lines were raised and maintained at 20 °C. For **a**, **e**, **i**, **j**, **k**, *n* = 100 nematodes per group. For **b**, **c**, **f**, **g**, *n* = 30 nematodes per group. For **d**, *n* = 20 (WT+control RNAi, WT+*let-363* RNAi or *clec-31*-OE+control RNAi) nematodes, *n* = 21 (*clec-31*-OE+*let-363* RNAi) nematodes. For **h**, *n* = 30 (*clec-31*-OE) nematodes, *n* = 29 (*clec-31*-OE+Rapamycin) nematodes. Three biologically independent experiments. Data are shown as mean ± s.d.; *P* values are derived from Log-rank (Mantel-Cox) test (**a**, **e**, **i**–**k**), one-way ANOVA with Dunnett's multiple comparisons test and Two-tailed unpaired Student's *t*-tests (**b**–**d**), Two-tailed unpaired Student's *t*-tests (**f**–**h**). Source data are provided as a Source data file.

containing a GST tag. STAT3 was cloned and inserted into pCMV-3xFlag for expression of the Flag tag fusion protein.

For the BiFC assay, the sequences of CD44ICD, PIK3R4 and STAT3 were cloned and inserted into both VN155 and VC155 (Addgene, MA, USA).

For the yeast two-hybrid assay, the sequences of ATG14L, BECN1, UVRAG, PIK3C3 and CD44ICD were cloned and inserted into pGADT7 as prey constructs. The pGADT7-PIK3R4 plasmid was purchased from Youbi Technology Co., Ltd. (Hunan, China). The sequences of STAT3 and CD44ICD were cloned and inserted into pGBKT7 as bait constructs.

For CD44 knockdown, a double-stranded RNA (shRNA) against CD44 was cloned and inserted into the pLKO.1-TRC cloning vector. The shRNA sequences targeting CD44 were as follows: F-5′-CCGGAAGTCTCAGGAAATGGTGCATCTCGAGATGCACCATTTCCTGA-GACTTTTTTTG-3′ and R-5′-AATTCAAAAAAAGTCTCAGGAAATGGTG-CATCTCGAGATGCACCATTTCCTGAGACTT-3′.

The LentiCRISPR v2 plasmid was used to construct the CRISPR-CD44 plasmid according to the manufacturer's instructions. Two sgCD44s were selected after verification of the effect of knockout by western blot analysis: sgCD44 forwards 1, CACCGTCGCTA-CAGCATCTCTCGGA; reverse 1, AAACTCCGAGAGATGCTGTAGCGAC; forwards 2, CACCGGAATACACCTGCAAAGCGGC; and reverse 2, AAACGCCGCTTTGCAGGTGTATTCC.

## Lentivirus preparation and transfection

HEK293T cells were co-transfected with 1 μg target plasmid, 250 ng PMD2.G and 750 ng PsPAX2 using Polyethylenimine Transfection Reagent. After 48 h, the medium containing lentivirus was collected and centrifuged at 6000 rpm for 30 min. The supernatant was filtered with a 0.45-μM filter, followed by centrifuging at 20,000 rpm for 2 h at 4 °C. HUVECs were grown to 70% confluency in 6-well plates and then subjected to infection using a lentivirus suspension that contained 4 μg/ml polybrene. Polybrene was diluted with 1 ml of fresh medium after 4 h. At 48 h posttransfection, fresh medium that contained 4 μg/mL puromycin was added, and the cells were incubated for another 24 h. The transfection efficiency was identified by western blot analysis.

## siRNA transfection

Specific siRNAs against CD44 and STAT3 and Stealth™ RNAi siRNA (scrambled siRNA) were purchased from Invitrogen. HUVECs were grown to 70% confluency in 6-well plates, washed and incubated in Opti-MEM. siRNAs (50 nM) and 7 μl of Lipofectamine RNAiMAX Reagent were mixed in 100 μl of Opti-MEM at room temperature for 10 min. The mixture was then added to the cells. After 6 h of incubation, the Opti-MEM was removed, and the cells were cultured for 24 h in normal culture medium supplemented with 10% FBS. The knockdown efficiency was confirmed by western blot analysis.

## Adenovirus transfection

HUVECs were transfected with siRNA or lentivirus according to the abovementioned methods. The medium was then changed to complete medium containing adenovirus expressing GFP-LC3B or mCherry-GFP-LC3 with a multiplicity of infection (MOI) of 10. The medium was changed 24 h after infection, and the corresponding treatments were then conducted.

## Ultrastructural analysis

After treatment, HUVECs and aortas were fixed overnight at 4 °C with 3% glutaraldehyde and then postfixed with 1% osmic acid for 2 h. The fixed samples were dehydrated using a graded ethanol series before being embedded in epoxy resin and sectioned. After staining with 2% uranyl acetate and lead citrate, the ultrathin sections were photographed under a JEM-1230 transmission electron microscope (JEOL Co., Ltd., Japan).

## PIK3C3 kinase activity assay

PIK3C3 activity was measured by quantitative and competitive ELISAs according to the manufacturer's manual (Echelon, K-3000). Briefly, PIK3C3 was immunoprecipitated using anti-ATG14L. The immunoprecipitates were incubated with kinase reaction buffer (20 mM Tris (pH 8.0), 200 mM NaCl, 2 mM EDTA, 20 mM MnCl$_2$, 100 μM ATP and phosphatidylinositol substrate (Echelon, K-3000)) and incubated at room temperature for 3 h. After the addition of the PtdIns3P detection buffer (provided by the kit, K-3004) and PtdIns3P detector protein (provided by the kit), the reaction mixture was added to the PtdIns3P-coated microplate. The amount of PtdIns3P detector protein bound to the plate was determined through colorimetric detection of the absorbance at 450 nm. The concentration of PtdIns3P in the reaction mixture was calculated as the inverse of the amount of PtdIns3P detector protein bound to the plate.

## Coimmunoprecipitation (IP)

After treatment, HUVECs were lysed in IP buffer (pH 7.5, 20 mM Tris, 137 mM NaCl, 1% NP-40, 5% glycerol, and 100 μM PMSF). Total cell lysates were incubated with anti-DDK or IgG overnight at 4 °C on a rocking platform. The immunocomplexes were incubated with protein A/G agarose beads for 1 h and then washed to remove the unbound immunocomplexes. The precipitates were separated by SDS–PAGE for further western blot analysis.

## Yeast two-hybrid screening

To validate the interaction between proteins, the bait and prey plasmids were cotransformed into the AH109 yeast strain using a yeast transformation system. The resulting prey and bait constructs were confirmed by sequencing analysis following the manufacturer's protocol. Transformants grown on yeast nitrogen base medium (YNB medium) lacking leucine and tryptophan (SD-Leu-Trp) were transferred to YNB medium lacking leucine, tryptophan, adenine and histidine (SD-Leu-Trp-Ade-His) and containing 5-bromo-4-chloro-3-indoxyl-α-D-galactopyranoside (X-α-gal). The known interacting protein pair p53 and SV40 large T-antigen was used as a positive control, and the human lamin protein, which does not interact with the SV40 large T-antigen, was included as a negative control.

## GST pull-down assay

The plasmids for GST-CD44ICD, GST-PIK3R4 and FLAG-STAT3 were transfected into BL21(DE3) cells and HUVECs. The fusion proteins were prepared according to the manufacturer's protocol. Approximately 100 μg of GST, GST-CD44ICD and GST-PIK3R4 fusion protein was immobilized in 50 μl of glutathione agarose and equilibrated before being incubated together at 4 °C for 2 h under gentle rocking. After 3–5 washes with PBST, approximately 100 μg of FLAG-STAT3 fusion protein was added to the immobilized GST, GST-CD44ICD and GST-PIK3R4. The two fusion proteins were incubated overnight at 4 °C under gentle rotation. The bound proteins were eluted with elution buffer (50 mM Tris, 150 mM NaCl, 10 mM GSH, pH 8.0) and analysed by western blot analysis.

## BiFC

Pairs of fusion plasmids were transferred into HUVECs for 36 h using the Lipofectamine 3000 Transfection Kit according to the manufacturer's protocol. The cells were then incubated at 25 °C for 8 h, fixed with 4% paraformaldehyde for 10 min, stained with DAPI, and then subjected to microscopy imaging. Cotransfection of VN-Jun and VC-Fos was used as a positive control.

## Prediction of the interaction between STAT3 and CD44ICD

After multiple-sequence alignment was performed with ClustalW V2.0.9, a hundred homology models of STAT3 and CD44ICD were built by Alphafold2[41]. The best model was selected for further refinement using the Chiron protein minimization server (http://chiron.dokhlab.org). The models were then verified by SAVES v6.0 (https://saves.mbi.ucla.edu/) and manually inspected with COOT[42]. The full-length protein binding modes of STAT3 and CD44ICD were studied using GRAMM, which is guided by the experimental details available for their interactions and functions. The protein–protein binding modes were studied and analysed using the PDBePISA server. The results were analysed using PyMOL 2.5.4 to characterize the critical amino acids in the protein–protein/peptide interaction interface.

## Heart rate, blood pressure and serum NO level measurements

The heart rate and systolic and diastolic blood pressures were non-invasively measured in conscious animals by the tail-cuff method using an intelligent noninvasive sphygmomanometer (Softron Biotechnology, BP-2010A). The animals were placed in a thermal insulation bucket, which was set to an appropriate temperature, wrapped in a rat bag, and recorded under steady-state conditions. The average of three consecutive blood pressure measurements was taken. For serum NO detection, the blood was allowed to rest for 1 h and then centrifuged for 10 min at $1000 \times g$ to achieve serum separation. The serum NO levels were determined using a nitric oxide assay kit according to the manufacturer's protocol.

## Vascular ultrasound and vascular permeability

Mice were anaesthetized with 5% chloral hydrate and examined by high-frequency ultrasound using a small animal ultrasound system (VINNO6 LAB). All recordings were made by a cardiac imaging technician blinded to the animal genotype. The diameters of the aortic arch and abdominal aorta lumen were monitored during systole and diastole. The blood flow velocity at the aortic arch was measured in pw mode. 100 μl of Evans Blue solution (0.5% Evans Blue Dye in PBS) was injected intravenously via retro-orbital injection into mice and allowed to circulate for 4 h, and the vascular permeability was determined by quantitative measurement of the dye incorporated per milligram of tissue (dry weight). Briefly, Evans blue dye was extracted from dried tissues by incubating the tissues in formamide at 65 °C overnight. The optical density of the extracted dye was measured at an absorbance of 620 nm using a spectrophotometer, and the concentration was calculated by comparison to the Evans blue dye standard curve and values normalized to the tissue dry weight.

## C. elegans and treatments

Wild-type nematodes (N2) were a gift from Prof. Ding Chunbang (Sichuan Agricultural University). The nematode strain DA2123 was a gift from Dr Zhang Hong (Institute of Biophysics, Chinese

Academy of Sciences). Transgenic nematodes overexpressing clec-31 (*clec-31* OE) were generated by SunyBiotech (Fuzhou, China). Briefly, the Peft-3 promoter region and *clec-31* genes were amplified from N2 nematode genomic DNA, and the linear vector ppd95.77 plasmid was digested by the Hind III and EcoR I enzymes. The two fragments were assembled together with the linear vector using Gibson assembly. Transformation was then conducted to select a single bacterium and extract the plasmid for sequencing to finally obtain the correct sequence. A mixture of 20 ng/μl plasmid DNA and 25 ng/μl reporter plasmid pTG964 was injected into the bilateral gonads of WT nematodes. Each transgenic nematode of the F1 generation was transferred into a single plate, and the free transgenic nematode was screened by a fluorescent reporter plasmid. The nematodes were maintained at 20 °C using nematode growth medium (NGM) agar plates with *E. coli* strain OP50 according to standard techniques. All animals used in this study are listed in Supplementary Table 2.

For RNAi experiments, *E. coli* HT115 strains harbouring plasmids expressing dsRNA against *C. elegans clec-31* and *let-363* genes were generated in our lab, and single-colony isolates were collected, purified and sequenced for the correct insert. Overnight cultures of *E. coli* HT115 bacteria containing vectors expressing dsRNA targeting genes or empty vector control (L4440) were grown in Luria–Bertani broth with 100 μg/ml ampicillin. The cultures were then concentrated and seeded onto NGM agar plates supplemented with 1 mM isopropyl β-D-1-thiogalactopyranoside (IPTG) and containing 100 μg/ml ampicillin. The seeded plates were incubated for at least 2 h at 25 °C before nematodes were transferred onto them. For rapamycin treatment, 50 mg/ml rapamycin was added to the agar plates at a concentration of 100 μM.

### Lifespan analysis
For nematodes, synchronized eggs were added to plates with *E. coli* strain OP50. On Day 1, animals were transferred to plates containing fresh control bacteria (containing empty vector L4440) or target RNAi bacteria or rapamycin daily to avoid mixed populations until they stopped laying eggs. The nematodes were then transferred every 2–3 days for the rest of the lifespan experiment. The nematode viability was scored every day. Death was defined by not responding to gentle tapping with a platinum wire. Nematodes that died from internal hatching, expulsion of internal contents or crawled off the plates were censored. The number of animals per test for each condition ranged from 100 to 120, whereas the valid number of animals was approximately 100. The lifespan analyses of nematodes were performed at 20 °C. For mice, 12–13 mice per genotype were allowed to live out their natural lifespan and were euthanized only if moribund.

### Motility, pharyngeal pumping, and lipofuscin assays
Synchronized eggs were added to plates with *E. coli* strain OP50. On Day 1, animals were transferred to plates containing fresh control bacteria, target RNAi bacteria or rapamycin daily. On Day 7, the number of body bends (defined as a change in direction of the bend at the mid-body) and pumps (defined as backwards grinder movements in the terminal bulb) per 30 s was counted under a stereomicroscope (M205, Leica, Wetzlar, Germany). Thirty nematodes per condition were used for each test. The motility of nematodes was recorded with a CCD video camera under a stereomicroscope. These experiments were performed at 20 °C. For the lipofuscin assay, nematodes were anaesthetized with 5 mM sodium azide and fixed on 2% agarose slides. Twenty to thirty nematodes per condition were used for each test. The blue autofluorescence of lipofuscin was observed under a fluorescence microscope. The autofluorescence intensity of each nematode was measured using ImageJ software.

### Data analysis and statistics
Data analyses were performed by unpaired two-tailed Student's *t* test or one-way ANOVA for multiple groups coupled with Tukey's multiple comparisons test or Dunnett's multiple comparisons test as indicated in the figure legends using GraphPad Prism 9 for Mac (Version 9.0.0). The log-rank Mantel–Cox test was used to compare the survival curves. All bar graphs show the mean values with error bars (±s.d.). Significance was considered if the *p* value was <0.05. The number of repeated experiments and the number of animals are indicated in the figure legends.

### Reporting summary
Further information on research design is available in the Nature Portfolio Reporting Summary linked to this article.

## Data availability
The authors declare that all data supporting the findings of this study are available within this paper and its Supplementary Files. The Source data are provided with this paper. The array data have been deposited into the Gene Expression Omnibus (GEO) database in National Center for Biotechnology Information (NCBI) under the accession GSE236523. Source data are provided with this paper.

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

## Acknowledgements

This work was funded by the National Natural Science Foundation of China (no. 32070742), the National Natural Science Foundation of Henan province (no. 232300421030), the Innovative Funds Plan of Henan University of Technology (no. 2021ZKCJ16) and the Double First-Class Discipline Construction Program of Henan University of Technology (nos. HAUTSYL2023KC05 and 0517-24410014). We thank Prof. Ding Chunbang (Sichuan Agricultural University) and Prof. Zhang Hong (Institute of Biophysics, Chinese Academy of Sciences) for providing nematodes.

## Author contributions

L.Z. defined the study concept and design; P.Y., Jx.C., Z.C., Zh.L., G.F., F.S., Zr.L., Z.X., Y.H., X.S., X.L., Jt.C., C.Z., P.F., L.C., Y.S., G.Z., H.J. and S.M. performed the experiments. L.Z., P.Y., Jx.C., Z.C., Zh.L., G.F., F.S. and Zr.L. performed the data analyses. L.Z. and P.Y. wrote the manuscript and assembled the figures; L.Z. provided facilities and material support; and all the authors have read and approved the final submitted paper.

## Competing interests
The authors declare no competing interests.
