## [Peer Review File · Nature Communications]

CD44 connects autophagy decline and ageing in the vascular endotheliumREVIEWER COMMENTS

Reviewer #1 (Remarks to the Author):

The manuscript by Zhang et al investigated the role of CD44 in autophagy decline in the vascular endothelium in the context of aging. The authors found that there is a direct link between CD44 and autophagy, presumably, by regulating PIK3R4 and PIK3C3 levels and STAT3-dependent disruption of PtdIns3K complexes in vascular endothelial cells.

This study proposed a potential mechanism of CD44-mediated autophagy decline in aging VECs. Previous studies established CD44 is highly expressed in senescent endothelial cells, herein, this study provides evidence of the mechanism by which CD44 reduces autophagy and its functional impact on VEC senescence. The concept of this manuscript has a certain novelty, and is indeed interesting as CD44ICD not only induced premature aging, but even shortened the lifespan of animals. However, I do have several concerns about the mechanism the authors proposed.

Major points:

1. This study observed the role and mechanism of CD44 on age-related autophagy decline and endothelial dysfunction, the latter significantly contributes to impaired microvascular perfusion, however, the outcomes about vascular physiology/vascular function in this study seem to be insufficient. One may wish to know if vascular function could be confirmed in vivo by performing measurements, for instance, Flow Mediated Dilatation, or vascular permeability.
2. There is strong evidence that advanced aging is associated with a progressive deterioration of microvascular homeostasis, at least in part, due to age-related impairment of angiogenic processes (PMID: 29795441, reference 3 in this manuscript). And importantly, CD44, as a well-known regulator in angiogenesis, generally promotes pathological angiogenesis in CVDs. Nevertheless, in this study, the authors found that CD44 induced premature aging and shortened the lifespan of animals. There are major inconsistencies that require addressing or discussion.
3. The authors analyze the expression level of LC3B-II rather than the ratio of LC3B-II/LC3B-I which reflects the level of autophagy more intuitively. Please provide the data.
4. In Extended Data Figure 1, though the author compared the endothelial autophagic activity in young CD44 WT and KO mice, the authors need to add groups to investigate the effect of CQ on the number of LC3B puncta and accumulation of SQSTM1 in the young endothelial cells here to consolidate the conclusion.
5. In Figure 2 g, the authors treated CD44-overexpressing HUVECs (showing decreasing autophagy) with bafilomycin A1 (an autophagy inhibitor). I am worried that the readers could not understand the purpose of this experiment adequately.

6. Please further strengthen the Extended Data Figure 8. Model for the role of CD44 links autophagy decline and aging in vascular endothelial cells. We expect insightful graphic abstract of this study. In particular, a figure reflecting autophagy detailedly and a better color scheme. This will increase readability.

Minor points:

1. In Figures 1g, 1m-1r, 2b, Extended Data Figure 1e, and so on, there are no scale bars in zoomed figures.
2. On Page 44, line 903, "g, Western blot analysis of C3B-II..." should be "g, Western blot analysis of LC3B-II ...".
3. In Figures 3a, 3b, and 3c, there are some mistakes about labeling group names "CD44CD44 Δ ICD" and "CD44CD44 Δ ECD". It seems like "CD44" is repeated.

Reviewer #2 (Remarks to the Author):

The age-associated decline of autophagy flux has been shown in some organs, but its mechanism is not entirely clear. Here the authors show an age-associated increase of cd44 in VECs, and that cd44-icd inhibits autophagy by disrupting PtdIns3K complexes in a stat3-dependent manner. Consistently, they find a premature aging phenotype in endothelium-specific CD44-ICD knock-in mice. They show some consistent phenotypes in *c. elegans*: shorter lifespan in *clec-31* (*lcd44* homologue) OE nematodes, and extended lifespan in *clec-31* RNAi nematodes. The cd44-stat3-autophagy axis appears to be a novel mechanism for the age-associated autophagy decline in VECs. Here are some questions for the authors' consideration.

An anti-autophagy activity of cytoplasmic STAT3 has been reported through a different mechanism (PMID: 23084476). The authors might want to comment on this.

Indeed, I am wondering if we can exclude the possibility that stat3 is involved in autophagy repression as a transcription factor. Indeed, the authors show that overexpression of cd44/icd enhances stat3-dependent transcription (ext fig 4d). In addition, cryptotanshinone inhibits stat3's transcription factor activities (although this is not a decisive experiment). Based on the data shown in ext fig 4b, stat1 is also affected. Does stat1 do the same as stat3 in this context? Stat1 and stat3 typically do the opposite as transcription factors.

Fig. 6m-q: they propose that the rapamycin/trehalose treatment alleviates cd44-icd-mediated senescence through autophagy activation. It is critical to confirm that autophagy is indeed activated in this condition. If cd44-icd disrupts the PtdIns3K complex, as proposed in this study, then I wouldn't expect that rapamycin rescues the autophagy reduction.

Along the same lines, they need to validate the activation of autophagy by mtor inhibition in clec-31 OE nematodes.

Also, in nematodes, it would be more feasible to directly test their hypothesis by knocking down autophagy genes in clec-31 RNAi as well as clec-31 OE+rapa conditions.

Other points:

Fig. 2a: a young control is missing. Although ext fig 1 shows a snapshot of young, it would be ideal to validate the flux in young as well.

line 281: why "unexpected?"

Reviewer #3 (Remarks to the Author):

The authors present a study describing the role and mechanisms that CD44 couples autophagy decline and aging in vascular endothelial cells. They showed increased CD44 that correlated with a decline in autophagy in aged vascular endothelium. The intercellular domain of CD44 (CD44ICD) reduced levels of PIK3R4 and PIK3C3 to decrease autophagy by disrupting the PtdIns3K complexes in STAT3 dependent manner. Moreover, knockdown of CD44 alleviated the age-associated phenotypes in both human and mouse ECs that show higher basal autophagy. EC-specific CD44ICD knock-in led to premature aging and reduced basal autophagy and decreased lifespan of the mice. Conversely, activation of autophagy reversed the premature aging of ECs with CD44ICD overexpression. CD44 homologue clec-31 was shown to have similar functions in *C. elegans*. Overall, this is an interesting study that reveals a mechanistic link between autophagy and aging in vascular endothelial cells. However, the following points need to be addressed:

1. Line 227-229, Besides PtdIns3k complex, did the authors examine the effect of CD44ICD on other autophagy machinery and/or regulators? A more systematic approach would improve this manuscript.

2. Figure 5c, the differences of PIK3R4 and PIK3C3 levels between two siCD44 and siSCR samples are very modest (indeed hard to see for PIK3C3). Could the authors test the effect on the enzymatic activities of these PtdIns3k complexes?

3. Figure 5 and Extended Figure 4 (lines 240-267), Could the authors provide a diagram to summarize data in these figures addressing the mechanisms of regulation of autophagy by CD44? Currently, it is very difficult for a reader to understand and fully make senses of these results. The authors showed that overexpression of CD44 or CD44ICD increased STAT3 transcription activity. They also concluded that CD44ICD decreases the levels of PIK3R4 and PIK3C3 and binds to these two core components through directly interacting with STAT3, thereby disrupting the formation of PtdIns3K complex and leading to reduced autophagy. So at least two different roles of STAT3 are involved (i.e. transcription and impact on PtdIns3K complex). Which is more important for regulation of autophagy activity? How does transcription activity of STAT3 affect autophagy in these cells?

Other points:

1. Figure 2e, 4th set of bars, are they CQ, or as shown in the image Bafilomycin A1?

2. Figure 2i is not described in the text.

3. Figures 3a and 3b, what is CD44CD44 Δ ICD?

4. Line 193, should "C-terminus" be N-terminus?

5. Figure 4d, please add blots for CD44 and CD44ICD.

6. Figure 4c, it will be helpful to add markers to structures that the authors like to highlight (e.g. autophagosomes).

7. Extended Data Figure 4a, please add p values for each gene.

Response to the Reviewers' Comments

Our deepest gratitude goes to the anonymous reviewers for their careful work and thoughtful suggestions that have helped improve this paper substantially.

Response to the reviewer 1's comments

Reviewer #1 (Remarks to the Author):

The manuscript by Zhang et al investigated the role of CD44 in autophagy decline in the vascular endothelium in the context of aging. The authors found that there is a direct link between CD44 and autophagy, presumably, by regulating PIK3R4 and PIK3C3 levels and STAT3-dependent disruption of PtdIns3K complexes in vascular endothelial cells. This study proposed a potential mechanism of CD44-mediated autophagy decline in aging VECs. Previous studies established CD44 is highly expressed in senescent endothelial cells, herein, this study provides evidence of the mechanism by which CD44 reduces autophagy and its functional impact on VEC senescence. The concept of this manuscript has a certain novelty, and is indeed interesting as CD44ICD not only induced premature aging, but even shortened the lifespan of animals. However, I do have several concerns about the mechanism the authors proposed.

Response: We thank the Reviewer#1 for providing constructive feedback and appreciate the positive comments highlighting the contributions of our study. We have performed additional experiments and revised our manuscript according to your suggestions.

Major

points:

1. This study observed the role and mechanism of CD44 on age-related autophagy decline and endothelial dysfunction, the latter significantly contributes to impaired microvascular perfusion, however, the outcomes about vascular physiology/vascular function in this study seem to be insufficient. One may wish to know if vascular function could be confirmed *in vivo* by performing measurements, for instance, Flow Mediated Dilation, or vascular permeability.

Response: Thank you very much for raising this important issue. We detected vascular physiology/vascular function *in vivo* according to your suggestions. The lumen diameters of the aortic arches and abdominal aortas and the blood flow velocity at the aortic arches of WT mice and CD44^{EC} KI mice were measured using a small animal ultrasound imaging system (VINNO 6 LAB, China). Compared with aortas from the age-matched control animals, aortas of CD44^{EC} KI mice had smaller lumen diameters during systole and diastole. CD44^{EC} KI mice showed higher blood flow velocity in the aortic arches, suggesting that overexpression of CD44 in VECs results to vascular lumen stenosis. These results were incorporated in the Results section (lines 298-300) and Fig. 6k-m in the revised manuscript.

In addition, we measured the basal vascular permeability of animals using the albumin-binding property of Evans blue dye. Evans blue dye was injected intravenously via retro-orbital injection into mice and allowed to circulate for 4 hours. At harvest, various organs were removed, weighed, and dried overnight. In the 7 different tissues that were evaluated (heart, liver, spleen, lung, spleen, ear, skin), there were differences

in the basal vascular permeability in the liver, lungs and skin, most likely as a consequence of impaired endothelial function. These results were incorporated in the Results section (lines 300-302) and Extended Data Fig. 6 in the revised manuscript.

2. There is strong evidence that advanced aging is associated with a progressive deterioration of microvascular homeostasis, at least in part, due to age-related impairment of angiogenic processes (PMID: 29795441, reference 3 in this manuscript). And importantly, CD44, as a well-known regulator in angiogenesis, generally promotes pathological angiogenesis in CVDs. Nevertheless, in this study, the authors found that CD44 induced premature aging and shortened the lifespan of animals. There are major inconsistencies that require addressing or discussion.

Response: Thank you very much for your valuable comments and providing us suggestions. In the present study, we uncovered the role of CD44 in promoting premature aging and shortening lifespan of animals. As mentioned by the reviewer, given that aging is at least partially due to impaired angiogenesis, our study seems to contradict previous findings that CD44 promotes pathologic angiogenesis. Our understanding of this problem is described below.

First, angiogenesis can be divided into two types, physiological and pathological. Pathways that are increasingly recognized as important in mediating pathological angiogenesis might not have an equally important role in developmental and other types of physiological angiogenesis. Although CD44 has been identified as an angiogenesis regulator in the pathological microenvironment, it is unclear whether it affects

angiogenesis under physiological conditions, including age-related decline in angiogenesis. In fact, it has been reported that CD44 deletion in mice leads to reduced pathological angiogenesis without affecting normal angiogenesis [1]. Some proteins exerted different, or even opposite, regulatory effects on angiogenesis in different environments. For example, SIRT6 promoted angiogenesis in plaque under hypoxia; on the other hand, SIRT6 promoted injury of neovascular under oxidative stress [2]. These results seem contradictory but can be well explained combined with the pathogenesis of CVDs. Another example is macrophage colony-stimulating factor (M-CSF), a cytokine required for pathological neovascularization but was not required for the recovery of normal vasculature [3]. In mouse osteosarcoma, M-CSF inhibition effectively suppressed tumor angiogenesis. Continuous M-CSF inhibition did not affect healthy vascular and lymphatic systems outside tumors [3]. The function of CD44 are highly dependent on the environment [4]. In our present study, we identified that CD44 is expressed at low levels in normal young HUVECs and mice vessels, which is consistent with the findings in the Human Protein Atlas database (<https://www.proteinatlas.org>): relative low expression of CD44 on cardiomyocytes and vascular endothelial cells in normal human heart tissue. Nevertheless, CD44 was upregulated in the myocardial infarcted and border zones [5]. Therefore, we realized that CD44 may not be necessary for angiogenesis under physiological conditions.

Second, recent evidence demonstrated that in contrast to healthy blood vessels, pathological neovascularization engaged pathways leading to p16 activation and ultimately culminating in VEC senescence [6]. Selective ablation of dysfunctional

neovascularization promotes regrowth of functional normal blood vessels [6]. These results led us to consider the possibility that increased CD44 levels in VECs may promote pathological angiogenesis, promoting VEC senescence and preventing physiological vascular repair. The observation that elevated CD44 levels are associated with plaque destabilization support this possibility [7]. This is an interesting issue that we will explore in the future studies. Based on these observations, our results are not inconsistent with previous findings. We have added a discussion on this important issue in the revised manuscript (lines 412-430).

Reference

- [1] Chen L, Fu C, Zhang Q, He C, Zhang F, Wei Q. The role of CD44 in pathological angiogenesis. *FASEB J.* 2020, 34(10):13125-13139.
- [2] Yang Z, Huang Y, Zhu L, Yang K, Liang K, Tan J, Yu B. SIRT6 promotes angiogenesis and hemorrhage of carotid plaque via regulating HIF-1 α and reactive oxygen species. *Cell Death Dis.* 2021, 12(1):77.
- [3] Kubota Y, Takubo K, Shimizu T, Ohno H, Kishi K, Shibuya M, Saya H, Suda T. M-CSF inhibition selectively targets pathological angiogenesis and lymphangiogenesis. *J Exp Med.* 2009, 206(5):1089-102.
- [4] Zhao L, Hall JA, Levenkova N, Lee E, Middleton MK, Zukas AM, Rader DJ, Rux JJ, Puré E. CD44 regulates vascular gene expression in a proatherogenic environment. *Arterioscler Thromb Vasc Biol.* 2007, 27(4):886-92.
- [5] Zhang Q, Chen L, Huang L, Cheng H, Wang L, Xu L, Hu D, He C, Fu C, Wei Q. CD44 promotes angiogenesis in myocardial infarction through regulating plasma

exosome uptake and further enhancing FGFR2 signaling transduction. *Mol Med.* 2022, 28(1):145.

- [6] Crespo-Garcia S, Tsuruda PR, Dejda A, Ryan RD, Fournier F, Chaney SY, Pilon F, Dogan T, Cagnone G, Patel P, Buscarlet M, Dasgupta S, Girouard G, Rao SR, Wilson AM, O'Brien R, Juneau R, Guber V, Dubrac A, Beausejour C, Armstrong S, Mallette FA, Yohn CB, Joyal JS, Marquess D, Beltran PJ, Sapiha P. Pathological angiogenesis in retinopathy engages cellular senescence and is amenable to therapeutic elimination via BCL-xL inhibition. *Cell Metab.* 2021, 33(4):818-832.e7.
- [7] Bot PT, Pasterkamp G, Goumans MJ, Strijder C, Moll FL, de Vries JP, Pals ST, de Kleijn DP, Piek JJ, Hoefler IE. Hyaluronic acid metabolism is increased in unstable plaques. *Eur J Clin Invest.* 2010, 40(9):818-27.

3. The authors analyze the expression level of LC3B-II rather than the ratio of LC3B-II/LC3B-I which reflects the level of autophagy more intuitively. Please provide the data.

Response: Thank you very much for your valuable comments. According to the reviewer's suggestion, we calculated LC3B-II/LC3B-I ratios and provided the data in our revised manuscript.

4. In Extended Data Figure 1, though the author compared the endothelial autophagic activity in young CD44 WT and KO mice, the authors need to add

groups to investigate the effect of CQ on the number of LC3B puncta and accumulation of SQSTM1 in the young endothelial cells here to consolidate the conclusion.

Response: Thank you very much for raising this important issue and providing us valuable suggestions. According the reviewer's suggestion, we evaluated autophagic flux in the endothelium of young animals injected with CQ. CQ increased the accumulation of LC3B and SQSTM1 in the endothelium of both young WT and CD44 KO mice. The level of LC3B in the endothelium was greater in CD44 KO mice than in WT mice with or without CQ, confirming that knockout of CD44 makes animals maintain a high level of basal autophagy in aging process. These results were incorporated in the Results section (lines 130-133) and Extended Data Fig. 1d and e in the revised manuscript.

5. In Figure 2 g, the authors treated CD44-overexpressing HUVECs (showing decreasing autophagy) with bafilomycin A1 (an autophagy inhibitor). I am worried that the readers could not understand the purpose of this experiment adequately.

Response: Thank you very much for your valuable comments. The decrease in the number of GFP-LC3B puncta and LC3B-II levels can reflect either decreased autophagic induction or increased autophagosome degradation. To discriminate between these two possibilities, we used bafilomycin A1 to block autophagosome

degradation. We added a sentence emphasizing the purpose of these experiments in our revised manuscript as below.

CD44-overexpressing HUVECs had decreased number of LC3B puncta, decreased levels of LC3B-II and an apparent accumulation of SQSTM1, which was not alter by bafilomycin A₁, suggesting that CD44 overexpression decreases autophagic activity rather than promotes autophagosome degradation.

6. Please further strengthen the Extended Data Figure 8. Model for the role of CD44 links autophagy decline and aging in vascular endothelial cells. We expect insightful graphic abstract of this study. In particular, a figure reflecting autophagy detailedly and a better color scheme. This will increase readability.

Response: Thank you very much for raising this important issue and providing us valuable suggestions. According the reviewer's suggestion, we designed a new graphical abstract to better reflect the function of CD44 linking vascular endothelial cell autophagy decline and aging. The new graphical abstract is presented as Extended Data Fig. 12 in the revised paper.

Minor

points:

1. In Figures 1g, 1m-1r, 2b, Extended Data Figure 1e, and so on, there are no scale bars in zoomed figures.

Response: Thank you very much for pointing this out and we added scale bars in all zoomed figures in our revised manuscript.

2. On Page 44, line 903, "g, Western blot analysis of C3B-II..." should be "g, Western blot analysis of LC3B-II ...".

Response: We feel sorry for our carelessness. We have corrected this mistake in our revised manuscript and we also feel great thanks for your point out.

3. In Figures 3a, 3b, and 3c, there are some mistakes about labeling group names "CD44CD44 Δ ICD" and "CD44CD44 Δ ECD". It seems like "CD44" is repeated.

Response: Thank you very much for pointing this out. The reviewer is correct, and we have corrected these mistakes in our revised manuscript.

Reviewer #2 (Remarks to the Author):

The age-associated decline of autophagy flux has been shown in some organs, but its mechanism is not entirely clear. Here the authors show an age-associated increase of cd44 in VECs, and that cd44-icd inhibits autophagy by disrupting PtdIns3K complexes in a stat3-dependent manner. Consistently, they find a premature aging phenotype in endothelium-specific CD44-ICD knock-in mice. They show some consistent phenotypes in c elegans: shorter lifespan in clec-31 (lcd44 homologue) OE nematodes, and extended lifespan in clec-31 RNAi nematodes. The cd44-stat3-autophagy axis appears to be a novel mechanism for the age-associated autophagy decline in VECs. Here are some questions for the authors' consideration.

Response: We thank the Reviewer #2 for your professional review work and constructive suggestions that greatly help us improve our manuscript. We have performed additional experiments and revised our manuscript according to your suggestions.

An anti-autophagy activity of cytoplasmic STAT3 has been reported through a different mechanism (PMID: 23084476). The authors might want to comment on this. Indeed, I am wondering if we can exclude the possibility that stat3 is involved in autophagy repression as a transcription factor. Indeed, the authors show that overexpression of cd44/icd enhances stat3-dependent transcription (ext fig 4d). In addition, cryptotanshinone inhibits stat3's transcription factor activities (although this is not a decisive experiment). Based on the data shown in ext fig 4b, stat1 is also affected. Does stat1 do the same as stat3 in this context? Stat1 and stat3 typically do the opposite as transcription factors.

Response: Thank you very much for raising this very important issue and providing us valuable suggestions. To test this idea, we generated a STAT3 variant that exclusively localizes to the cytoplasm due to the addition of a nuclear export sequence (NES). Autophagy repression by STAT3 persisted when we employed NES-STAT3, excluding the possibility that STAT3 is involved in autophagy decline as a transcription factor. CD44/CD44ICD functions as a negative regulator of autophagy in a STAT3 transcription-independent manner. These results were incorporated in the Results section (lines 257-263) and Extended Data Fig. 5j in the revised manuscript.

Of note, nonphosphorylatable STAT3 mutant STAT3^{Y705F} failed to repress autophagy in HUVECs. These results were incorporated in the Results section (lines 257-263) and Extended Data Fig. 5i in the revised manuscript. This is consistent with the results that inhibition of STAT3 Tyr705 phosphorylation by cryptotanshinone can reverse the inhibitory effect of CD44ICD on autophagy. Cryptotanshinone significantly inhibits STAT3-dependent luciferase activity, the STAT3 Tyr705 phosphorylation and the dimerization of STAT3. Our results collectively demonstrated that promoting the phosphorylation of STAT3 at Tyr705 rather than STAT3 transcription is necessary for CD44/CD44ICD to repress autophagy.

Actually, we present several important pieces of evidence showing that p-STAT3 is an important regulator of autophagy. First, knockdown of CD44 reduced p-STAT3^{Tyr705} level and increased LC3-II/LC3-I ratio, whereas overexpression CD44 or CD44ICD increased p-STAT3^{Tyr705} level and decreased LC3-II/LC3-I ratio, implying the association of p-STAT3 and autophagy. Second, a strong antagonist of p-STAT3 at Tyr705 effectively reversed the inhibitory effect of CD44ICD on autophagy. Third, when the mutant STAT3^{Y705F} was transfected into HUVECs, it did not function as an autophagic inhibitor. However, a previous report has shown that STAT3 inhibits autophagy *via* a cytoplasmic mechanism that does not involve the phosphorylation of Y705 [1]. Obviously, these results are conflicting and may be explained by the different cell types used. Qin *et al.* reported that, unlike wild-type STAT3, overexpression of STAT3^{Y705F} failed to repress autophagy in human leukemic monocyte lymphoma cell line U937 cells, providing evidence that p-STAT3 mediates the inhibition of starvation-

induced autophagy by IL-6 [2]. These findings support our observations. We have added a discussion on this important issue in the revised manuscript (lines 377-387).

In addition, we examined whether STAT1 changes with CD44 at the protein level. Our results showed that, unlike STAT3, the changes of STAT1 were inconsistent with the changes of CD44. That is, the expression levels of STAT1 and p-STAT1/STAT1 were decreased regardless of whether CD44 (or CD44ICD) was up-regulated or down-regulated. Therefore, it is unlikely that STAT1 plays the same role as STAT3 in CD44-regulated autophagy. These data are shown as below.

Reference

- [1] Shen S, Niso-Santano M, Adjemian S, Takehara T, Malik SA, Minoux H, Souquere S, Mariño G, Lachkar S, Senovilla L, Galluzzi L, Kepp O, Pierron G, Maiuri MC, Hikita H, Kroemer R, Kroemer G. Cytoplasmic STAT3 represses autophagy by inhibiting PKR activity. *Mol Cell*. 2012, 48(5):667-80.
- [2] Qin B, Zhou Z, He J, Yan C, Ding S. IL-6 Inhibits Starvation-induced Autophagy via the STAT3/Bcl-2 Signaling Pathway. *Sci Rep*. 2015, 5:15701.

Fig. 6m-q: they propose that the rapamycin/trehalose treatment alleviates cd44-icd-mediated senescence through autophagy activation. It is critical to confirm that autophagy is indeed activated in this condition. If cd44-icd disrupts the PtdIns3K complex, as proposed in this study, then I wouldn't expect that rapamycin rescues the autophagy reduction. Along the same lines, they need to validate the activation of autophagy by mtor inhibition in clec-31 OE nematodes. Also, in nematodes, it would be more feasible to directly test their hypothesis by knocking down autophagy genes in clec-31 RNAi as well as clec-31 OE+rapa conditions.

Response: Thank you very much for your raising this important issue and providing us valuable suggestions. We provided evidence that rapamycin or trehalose activates autophagy in VECs overexpressing CD44ICD *in vitro* and *in vivo*. A significant increase in the LGG-1-II/LGG-1-I ratio suggested that autophagy was increased in *clec-31* OE nematodes fed with bacteria expressing mTOR/let-363 dsRNA or treated with

rapamycin. These results were incorporated in our revised paper as Extended Data Fig. 8 and Extended Data Fig. 11a.

According to the reviewer's suggestions, we knocked down autophagy genes in *clec-31* RNAi as well as *clec-31* OE+rapa conditions. The results showed that genetic inhibition of autophagy by RNAi against *unc-51* or *vps34* suppressed lifespan extension of the *clec-31* knockdown nematodes and rapamycin-treated *clec-31* OE nematodes, confirming that *clec-31* accelerates aging in *C. elegans* by reducing autophagy. These results were incorporated in the Results section (lines 350-352) and Fig. 7u-w and Extended Data Fig. 11b in the revised manuscript.

As mentioned by the reviewer, it seems incomprehensible that rapamycin rescues CD44ICD-induced autophagy reduction. In fact, many studies have demonstrated that rapamycin inhibits the expression and phosphorylation of STAT3 [1-5]. A recent study confirmed that STAT3 is a direct target of rapamycin [6]. We also found that rapamycin decreased the level of STAT3 and p-STAT3 in CD44ICD-overexpressing HUVECs. These data are shown as below.

Figure legends Western blot analysis of CD44ICD-overexpressing HUVECs treated with DMSO

or rapamycin. Data (n = 3 independent experiments) are shown as mean \pm s.d.; *P* values are derived from Two-tailed unpaired Student's *t*-tests.

Of note, nonphosphorylatable STAT3 mutant STAT3^{Y705F} failed to repress autophagy in HUVECs. These results were incorporated in the Results section (lines 257-263) and Extended Data Fig. 5i in the revised manuscript. Considering that CD44ICD functions through p-STAT3, it is not difficult to understand why rapamycin could rescue CD44ICD-induced autophagy repression.

Moreover, we built the STAT3 and CD44ICD models by Alphafold2, and then the protein-protein binding modes were studied and analyzed by using GRAMM-X and PDBePISA server. The affinity energy of the STAT3-CD44ICD complex is -7.2 kcal/mol, which indicates they bind tightly. The binding stability of the STAT3-CD44ICD complex is supported by the N-terminal 35 amino acids of CD44ICD, which formed multiple hydrogen bonds between STAT3 and CD44ICD. From the Co-IP analysis, CD44ICD missing 35 amino acids at the N-terminal (CD44ICD_ΔN35) failed to bind STAT3, PIK3R4 and PIK3C3, indicating that the 35 amino acids are necessary for the combination of CD44ICD and STAT3. Interestingly, the disability to inhibit autophagy of mutant CD44ICD_ΔN35 was observed in our data (Fig. 3m), which is supportive of the viewpoint that CD44ICD blocks autophagy via interacting with STAT3. These results were incorporated in the Results section (lines 273-284) and Fig. 5l and m in the revised manuscript.

Reference

[1] Wang R, Yu Z, Sunchu B, Shoaf J, Dang I, Zhao S, Caples K, Bradley L, Beaver

- LM, Ho E, Löhr CV, Perez VI. Rapamycin inhibits the secretory phenotype of senescent cells by a Nrf2-independent mechanism. *Aging Cell*. 2017, 16(3):564-574.
- [2] Zhou J, Wulfschlegel J, Zhang H, Gu P, Yang Y, Deng J, Margolick JB, Liotta LA, Petricoin E 3rd, Zhang Y. Activation of the PTEN/mTOR/STAT3 pathway in breast cancer stem-like cells is required for viability and maintenance. *Proc Natl Acad Sci U S A*. 2007, 104(41):16158-63.
- [3] Hou H, Miao J, Cao R, Han M, Sun Y, Liu X, Guo L. Rapamycin Ameliorates Experimental Autoimmune Encephalomyelitis by Suppressing the mTOR-STAT3 Pathway. *Neurochem Res*. 2017, 42(10):2831-2840.
- [4] Mao X, Cho MJT, Ellebrecht CT, Mukherjee EM, Payne AS. Stat3 regulates desmoglein 3 transcription in epithelial keratinocytes. *JCI Insight*. 2017, 2(9):e92253.
- [5] Hong SM, Park CW, Cha HJ, Kwon JH, Yun YS, Lee NG, Kim DG, Nam HG, Choi KY. Rapamycin inhibits both motility through down-regulation of p-STAT3 (S727) by disrupting the mTORC2 assembly and peritoneal dissemination in sarcomatoid cholangiocarcinoma. *Clin Exp Metastasis*. 2013, 30(2):177-87.
- [6] Sun L, Yan Y, Lv H, Li J, Wang Z, Wang K, Wang L, Li Y, Jiang H, Zhang Y. Rapamycin targets STAT3 and impacts c-Myc to suppress tumor growth. *Cell Chem Biol*. 2022, 29(3):373-385.e6.

Other

points:

Fig. 2a: a young control is missing. Although ext fig 1 shows a snapshot of young, it would be ideal to validate the flux in young as well.

Response: We agree to this point. According to the reviewer's suggestion, we evaluated the autophagic flux in the endothelium of young animals injected with CQ. CQ increased the accumulation of LC3B and SQSTM1 in the endothelium of both young WT and CD44 KO mice. The level of LC3B in the endothelium was greater in CD44 KO mice than in WT mice with or without CQ, confirming that knockout of CD44 makes animals maintain a high level of basal autophagy in aging process. These results were incorporated in the Results section (lines 130-133) and Extended Data Fig. 1d and e in the revised manuscript.

line 281: why "unexpected?"

Response: Thank you very much for pointing this out. We originally wanted to express that "The fact that overexpression of CD44ICD in the vascular endothelium alone could affect the lifespan of mice is more than our expectation". We changed "unexcepted" to "more than our expectation" in our revised manuscript.

Reviewer #3 (Remarks to the Author):

The authors present a study describing the role and mechanisms that CD44 couples autophagy decline and aging in vascular endothelial cells. They showed increased CD44 that correlated with a decline in autophagy in aged vascular

endothelium. The intercellular domain of CD44 (CD44ICD) reduced levels of PIK3R4 and PIK3C3 to decrease autophagy by disrupting the PtdIns3K complexes in STAT3 dependent manner. Moreover, knockdown of CD44 alleviated the age-associated phenotypes in both human and mouse ECs that show higher basal autophagy. EC-specific CD44ICD knock-in led to premature aging and reduced basal autophagy and decreased lifespan of the mice. Conversely, activation of autophagy reversed the premature aging of ECs with CD44ICD overexpression. CD44 homologue clec-31 was shown to have similar functions in *C. elegans*. Overall, this is an interesting study that reveals a mechanistic link between autophagy and aging in vascular endothelial cells. However, the following points need to be addressed:

Response: We thank the Reviewer #3 for providing constructive feedback and appreciate the positive comments highlighting the contributions of our study. We have performed additional experiments and revised our manuscript according to your suggestions.

1. Line 227-229, Besides PtdIns3k complex, did the authors examine the effect of CD44ICD on other autophagy machinery and/or regulators? A more systematic approach would improve this manuscript.

Response: Thank you very much for raising this important issue and providing us valuable suggestions. In fact, since CD44 inhibits autophagy induction, in addition to the PtdIns3k complex, we also examined the ULK1 complex, both of which are play

central roles for initiation of autophagy. A range of signaling processes converge on two protein complexes to initiate autophagy: the ULK1 protein kinase complex and the PtdIns3k kinase complex. Although knockdown of CD44 increased the levels of three core components of ULK1 complex, including ULK1, ATG13 and ATG101, no changes in the levels of the core components of ULK1 complex were detected in CD44ICD-overexpressing HUVECs. Therefore, we focused our research on the impact of CD44ICD on the PtdIns3k complex. These results were incorporated in the Results section (lines 223-229) and Extended Data Fig. 4 in the revised manuscript.

2. Figure 5c, the differences of PIK3R4 and PIK3C3 levels between two siCD44 and siSCR samples are very modest (indeed hard to see for PIK3C3). Could the authors test the effect on the enzymatic activities of these PtdIns3k complexes?

Response: Thank you very much for raising this important issue and providing us valuable suggestions. A PIK3C3 kinase activity assay was performed according to the reviewer's suggestion. After an immunoprecipitation (IP) assay in HUVECs transduced with empty vector, CD44 or CD44ICD with anti-ATG14L antibody was carried out, we incubated the anti-ATG14L immunoprecipitate with phosphatidylinositol as the substrate and conducted quantitative ELISA. The results revealed that PtdIns3K kinase activity was significantly decreased in CD44/CD44ICD-overexpressing HUVECs. These results were incorporated in the Results section (lines 240-243) and Fig. 5e in the revised manuscript.

3. Figure 5 and Extended Figure 4 (lines 240-267), Could the authors provide a diagram to summarize data in these figures addressing the mechanisms of regulation of autophagy by CD44? Currently, it is very difficult for a reader to understand and fully make senses of these results. The authors showed that overexpression of CD44 or CD44ICD increased STAT3 transcription activity. They also concluded that CD44ICD decreases the levels of PIK3R4 and PIK3C3 and binds to these two core components through directly interacting with STAT3, thereby disrupting the formation of PtdIns3K complex and leading to reduced autophagy. So at least two different roles of STAT3 are involved (i.e. transcription and impact on PtdIns3K complex). Which is more important for regulation of autophagy activity? How does transcription activity of STAT3 affect autophagy in these cells?

Response: Thank you very much for your raising this important issue and providing us valuable suggestions. As suggested by the reviewers, we provided a diagram to reflect the mechanism of CD44/CD44ICD in regulating autophagy. The diagram is presented as Fig. 5n in the revised paper.

As mentioned by the reviewer, we showed that overexpression of CD44 or CD44ICD increased STAT3 transcription activity. CD44ICD disrupts the formation of PtdIns3K complex by directly interacting with STAT3, leading to reduced autophagy. Which is more important for regulation of autophagy activity? How does transcription activity of STAT3 affect autophagy in these cells? To answer these questions, we generated a STAT3 variant that exclusively localizes to the cytoplasm due to the addition of a

nuclear export sequence (NES). Autophagy repression by STAT3 persisted when we employed NES-STAT3, excluding the possibility that STAT3 is involved in autophagy decline as a transcription factor. Although CD44/CD44ICD increased STAT3 transcription activity, CD44/CD44ICD functions as a negative regulator of autophagy in a STAT3 transcription-independent manner. These results were incorporated in the Results section (lines 257-263) and Extended Data Fig. 5j in the revised manuscript.

Moreover, we built the STAT3 and CD44ICD models by Alphafold2, and then the protein-protein binding modes were studied and analyzed by using GRAMM-X and PDBePISA server. The affinity energy of the STAT3-CD44ICD complex is -7.2 kcal/mol, which indicates they bind tightly. The binding stability of the STAT3-CD44ICD complex is supported by the N-terminal 35 amino acids of CD44ICD, which formed multiple hydrogen bonds between STAT3 and CD44ICD. From the Co-IP analysis, CD44ICD missing 35 amino acids at the N-terminal (CD44ICD_ΔN35) failed to bind STAT3, PIK3R4 and PIK3C3, indicating that the 35 amino acids are necessary for the combination of CD44ICD and STAT3. Interestingly, the disability to inhibit autophagy of mutant CD44ICD_ΔN35 was observed in our data (Fig. 3m), which is supportive of the viewpoint that CD44ICD blocks autophagy via interacting with STAT3. These results were incorporated in the Results section (lines 273-284) and Fig. 5l and m in the revised manuscript.

Other

points:

1. Figure 2e, 4th set of bars, are they CQ, or as shown in the image Bafilomycin

A1?

Response: We feel sorry for our carelessness. This is an error, we have changed “CQ” to “Bafilomycin A₁” in our revised manuscript.

2. Figure 2i is not described in the text.

Response: We feel sorry for our carelessness. We have corrected this mistake in our revised manuscript and we also feel great thanks for your point out.

Moreover, CD44-overexpressing HUVECs exhibited decreased number of both autophagosomes and autolysosomes compared with control cells (Fig. 2i and j), collectively demonstrating that CD44 decreases autophagy in VECs.

3. Figures 3a and 3b, what is CD44CD44ΔICD?

Response: Thank you very much for your comments. This is a writing error, we changed “CD44CD44ΔICD” to “CD44ΔICD” in our revised manuscript.

4. Line 193, should “C-terminus” be N-terminus?

Response: Thank you very much for pointing this out and we corrected this mistake in our revised manuscript.

5. Figure 4d, please add blots for CD44 and CD44ICD.

Response: Thank you very much for your comments. We added blots for CD44 and CD44ICD in Figure 4d.

6. Figure 4c, it will be helpful to add markers to structures that the authors like to highlight (e.g. autophagosomes).

Response: Thank you very much for your comments. According to the reviewer's suggestion, we added asterisks to indicate autophagosomes/autolysosomes in Figure 4c and 4g.

7. Extended Data Figure 4a, please add p values for each gene.

Response: Thank you very much for your comments. We added *p* values for STATs in Extended Data Figure 5a. "Extended Data Figure 4" is changed to "Extended Data Figure 5" in the revised manuscript.

REVIEWERS' COMMENTS

Reviewer #1 (Remarks to the Author):

The authors addressed my concerns and I have no more questions.

Reviewer #2 (Remarks to the Author):

The authors have addressed my questions adequately.

Reviewer #3 (Remarks to the Author):

The authors have addressed all my previous concerns. The manuscript is significantly improved. I don't have any other concern for the publication of the revised manuscript.

Response to the Reviewers' Comments

Our deepest gratitude goes to the anonymous reviewers for their careful work and comments.

Reviewer #1 (Remarks to the Author):

The authors addressed my concerns and I have no more questions.

Our response: We thank the reviewer very much for the comment.

Reviewer #2 (Remarks to the Author):

The authors have addressed my questions adequately.

Our response: We thank the reviewer very much for the comment.

Reviewer #3 (Remarks to the Author):

The authors have addressed all my previous concerns. The manuscript is significantly improved. I don't have any other concern for the publication of the revised manuscript.

Our response: We thank the reviewer very much for the comment.